# Distance Adaptive Beam Search for Provably Accurate Graph-Based Nearest Neighbor Search

**Yousef Al-Jazzazi**
New York University
Abu Dhabi
ya2225@nyu.edu

**Haya Diwan**
New York University
hd2371@nyu.edu

**Jinrui Gou**
New York University
jg6226@nyu.edu

**Cameron Musco**
UMass Amherst
cmusco@cs.umass.edu

**Christopher Musco**
New York University
cmusco@nyu.edu

**Torsten Suel**
New York University
torsten.suel@nyu.edu

## Abstract

Nearest neighbor search is central in machine learning, information retrieval, and databases. For high-dimensional datasets, graph-based methods such as HNSW, DiskANN, and NSG have become popular thanks to their empirical accuracy and efficiency. These methods construct a directed graph over the dataset and perform beam search on the graph to find nodes close to a given query. While significant work has focused on practical refinements and theoretical understanding of graph-based methods, many questions remain. We propose a new distance-based termination condition for beam search to replace the commonly used condition based on beam width. We prove that, as long as the search graph is *navigable*, our resulting Adaptive Beam Search method is guaranteed to approximately solve the nearest-neighbor problem, establishing a connection between navigability and the performance of graph-based search. We also provide extensive experiments on our new termination condition for both navigable graphs and approximately navigable graphs used in practice, such as HNSW and Vamana graphs. We find that Adaptive Beam Search outperforms standard beam search over a range of recall values, data sets, graph constructions, and target number of nearest neighbors. It thus provides a simple and practical way to improve the performance of popular methods.

## 1 Introduction

High-dimensional nearest neighbor search is a basic building block in many areas, including image and video processing [18, 26], information retrieval [6, 51], and algorithm design [10, 28]. It is central to modern machine learning, underlying document and media search based on learned embeddings [9, 40, 48], as well as retrieval-augmented generation (RAG) systems for large language models [37, 46]. Nearest neighbor search also plays a role in hard-negative mining [62], accelerating transformer architectures [29], and other applications across machine learning [58].

Formally, in the $k$-nearest neighbor search problem, we are given a set of data points, often machine-learned vector embeddings of documents, images, or other media [12, 14]. We are also given a distance measure, such as the Euclidean distance, or something more exotic like Chamfer distance [25]. The goal is to pre-process the dataset into a search data structure so that, given any query point $q$, we can efficiently find the $k$ data points closest to $q$ with respect to the distance measure.

Solving this problem exactly is notoriously difficult in high dimensions, so applications typically rely on approximate nearest neighbor (ANN) methods that attempt to find *most* of the $k$ closest neighbors.

39th Conference on Neural Information Processing Systems (NeurIPS 2025).

Popular ANN methods include locality sensitive hashing (LSH) [2, 3, 23, 41], inverted file indices based on quantization or clustering [26, 27, 49], and more [8, 30, 34]. In this work, we focus on *graph-based* ANN methods, which have been extensively studied and perform extremely well in practice, topping leader boards at several recent ANN competitions [55, 56].

**Graph-Based Nearest Neighbor Search.** The high-level idea of graph-based methods is simple. We construct an index by building a directed graph, $G$, with one node for each data point. Given a query, $q$, we search the index by starting at an arbitrary node and performing a greedy graph traversal, exploring neighbors in that graph that are closest to $q$. A specific choice of graph construction and traversal method comprises a particular "graph-based" nearest neighbor search method.

Many algorithms for graph construction have been proposed, including the Hierarchical Navigable Small World (HNSW) approach [43], Vamana/DiskANN [33, 59], Navigating Spreading-out Graphs (NSG) [16], and others [45, 60]. All of these methods construct a graph $G$ that, for a given node $i$, contains out-edges to nearest neighbors of $i$, as well as "long range" connections to nodes far away from $i$. Such constructions are loosely motivated by the concept of *navigability*, which dates back to pioneering work on local graph routing by Kleinberg [31, 32] and Milgram [47]. We provide a formal definition of navigability in Section 2, but the property roughly guarantees that there is a path from any node $i$ in $G$ to any node $j$ so that the distance to $j$ strictly decreases along the path.

While graph constructions vary greatly, the choice of greedy traversal method used in graph-based nearest neighbor search has seen less innovation. A variant of greedy search called *beam search* is almost ubiquitous. Parameterized by a beam width $b \geq k$, beam search maintains a list of $b$ candidate nearest neighbors and computes the query's distance to each of those candidates' neighbors, updating them until it fails to find any better candidates. See Section 3 for a formal description.

**Our Contributions.** While graph-based ANN methods have seen significant practical success, their performance is poorly understood from a theoretical perspective. This is in contrast to methods like locality sensitive hashing, for which it is possible to prove strong worst-case approximation guarantees [2, 4]. A lack of theory makes it difficult to iterate on and improve existing graph-based methods, and to understand the limitations of these methods. We aim to address this theory-practice gap, and in turn, introduce principled improvements to existing methods.

In particular, we re-examine the ubiquitous beam search method, viewing it, as in some previous work [17], as a specific stopping rule for a much more general search procedure. This perspective motivates a new algorithm called Adaptive Beam Search, which stops searching for candidates based on a *distance-based criterion* instead of a fixed beam width, $b$. Our main theoretical result (Theorem 1) is to prove that Adaptive Beam Search returns provable approximate nearest neighbors *whenever the search graph $G$ is navigable*. To the best of our knowledge, this result is the first to theoretically connect the performance of greedy search (specifically, beam search) to the property of navigability.

Moreover, our theoretical results translate into practical performance. We perform an extensive experimental evaluation of Adaptive Beam Search, comparing it to fixed-width beam search over a wide range of data sets, graph constructions, recall values, and target number of nearest neighbors. The method universally outperforms classic beam search, typically providing a $10 - 50\%$ reduction in the number of distance computations required for a given level of recall. Moreover, Adaptive Beam Search can be implemented with only minor code changes to existing graph-based libraries. We thus hope that, beyond its theoretical relevance, the method will have practical impact.

**Roadmap.** The remainder of this paper is organized as follows. In Section 2, we discuss technical preliminaries and related work. In Section 3, we introduce our Adaptive Beam Search method and its motivating ideas. In Section 4, we prove that Adaptive Beam Search solves approximate nearest neighbor search on navigable graphs (Theorem 1). In Section 5, we evaluate Adaptive Beam Search on sparse navigable graphs and common heuristic graph constructions, including HNSW and Vamana.

## 2 Background and Related Work

We start by defining notation used throughout. Our goal in this paper is to find nearest neighbors in a metric space $\mathcal{X}$ equipped with a distance function $d : \mathcal{X} \times \mathcal{X} \to \mathbb{R}^+$.[1] We are given a database

---

[1]In particular, we just require that for all $i, j, k \in \mathcal{X}$, $d(i, j) = d(j, i)$, $d(i, j) > 0$ when $i \neq j$, $d(i, i) = 0$, and $d(i, j) + d(j, k) \geq d(i, k)$ (i.e., triangle inequality holds).

of $n$ items in $\mathcal{X}$, which we label $\{1, \ldots, n\}$. We want to find the nearest $k \leq n$ items to a given query $q \in \mathcal{X}$. E.g., for $k = 1$, the goal is to find $\operatorname{argmin}_{j \in \{1, \ldots, n\}} d(q, j)$. To avoid corner cases, we assume items in the database are unique, i.e., $d(i, j) > 0$ for all $i, j \in \{1, \ldots, n\}$, $i \neq j$.

In practice, the $n$ database items and the query $q$ are usually associated with vectors (e.g., machine learned embeddings) $\mathbf{x}_1, \ldots, \mathbf{x}_n$ and $\mathbf{x}_q \in \mathbb{R}^m$. The distance function $d(i, j)$ is chosen to be some function of these vectors, e.g., the Euclidean distance, $d(i, j) = \|\mathbf{x}_i - \mathbf{x}_j\|_2$.

**Graph Navigability.** Our theoretical guarantees assume use of a *navigable search graph* over $n$ nodes corresponding to our $n$ database items. While the term "navigable" is sometimes used informally in the literature, we use the following precise definition. Consider a directed graph $G = (V, E)$, with $V = \{1, \ldots, n\}$. For a node $x$, let $\mathcal{N}_G(x)$ denote its set of *out-neighbors*. Define:

**Definition 1** (Navigable Graph). *A directed graph $G$ is* navigable *under distance function $d$ if for any nodes $x, y \in \{1, \ldots, n\}$ with $d(x, y) > 0$, there is some $z \in \mathcal{N}_G(x)$ with $d(z, y) < d(x, y)$.*

Navigability ensures that, for any starting node $s$ and target node $t$, a standard greedy search where we always move to the neighbor of the current node closest to $t$, always converges to $t$.

When all distances between $\{1, \ldots, n\}$ are unique (this can be ensured by simply tie-breaking based on node id) it was recently shown that *any data set* has an efficiently computable navigable graph with average degree $O(\sqrt{n})$ for *any distance metric* [13, 11]. While the above bound is nearly optimal for worst-case data sets, much sparser navigable graphs often exist. For the Euclidean distance in $m$ dimensions, Arya and Mount construct navigable graphs with degree $2^{O(m)}$ [5]. For general metrics, Indyk and Xu construct navigable graphs with degree $2^{O(m') \log \Delta}$ where $m'$ is the doubling dimension of the data under $d$ and $\Delta = \max_{i,j} d(i, j) / \min_{i,j} d(i, j)$ is the dynamic range [24].

Why do we focus on navigability? Navigability has become a standard notion of "quality" for graphs used in nearest neighbor search. Indeed, the term lends its name to popular graph-based search methods such as the Navigable Small World (NSW) [42] and Hierarchical Navigable Small World (HNSW) [43] methods. Neither of these methods constructs graphs that are *provably* navigable, although they produce graphs that should be approximately navigable in practical settings. Surprisingly, however, to the best of our knowledge, no prior work formally links the accuracy of graph-based search to this intuitive notion of graph quality. As discussed, a major goal here is to address this theory-practice gap, and to use the resulting theory to propose new practical algorithms.

Related to our approach is a recent paper by Indyk and Xu [24] (and more recent follow-up work [20]) which proves accuracy guarantees for standard beam search under the assumption that the search graph is "$\alpha$-shortcut reachable", a strictly stronger criterion than navigability. A graph is $\alpha$-shortcut reachable if, for all $x, y \in \{1, \ldots, n\}$ with $d(x, y) > 0$, there is some $z \in \mathcal{N}_G(x)$ with $\alpha \cdot d(z, y) < d(x, y)$ for some parameter $\alpha \geq 1$. Indeed, navigability exactly corresponds to this definition with $\alpha = 1$. However, the results from [24, 20] only yield a bounded approximation factor for $\alpha > 1$ (concretely, [24] obtains approximation factor $\frac{1+\alpha}{1-\alpha}$). Thus, obtaining theoretical results for graphs that are simply navigable remains an open question.

One reason this question is of practical importance is that navigable graphs can in general be much sparser than $\alpha$-shortcut reachable graphs. While it is possible to construct a navigable graph with average degree $O(\sqrt{n})$ for any database under any metric (under the mild assumption of unique distances) [11], it is not hard to observe that for any fixed $\alpha > 1$, even a random point set in $O(\log n)$-dimensional Euclidean space does not admit an $\alpha$-shortcut reachable graph with average degree $< n - 1$ with high probability (see Appendix A.1 for details). This is also the case for other stronger versions of navigability studied in recent work, like "$\tau$-monotonicity" [52, 22]

## 2.1 Additional Related Work

Beyond [24, 20], a few other papers have studied graph-based ANN search from a theoretical perspective. E.g., [35] and [54] study time-space tradeoffs akin to those available for LSH methods, but only for random data. More significant work has focused on practical algorithmic improvements. E.g., work has studied parallel implementations [45], methods for dynamic datasets [57, 63], distance approximations [64], graph pruning [65], filtered search [19], search with coverage criteria [1], and better search initialization strategies [66]. There has been relatively little work on alternatives to beam width-based termination in beam search, although a few papers study "early stopping" criteria that

incorporate distance information like our Adaptive Beam Search [44, 21]. There has also been some work on using machine learning to predict an optimal termination point [38]. In concurrent work, [61] propose another algorithm called *Adaptive Beam Search*, but the approach is fundamentally different than ours: they adapt the edges considered during the search process based on the query $q$. They give theoretical bounds assuming $G$ is $\tau$-monotonic, a stronger variant of navigability [52, 22].

## 3 Adaptive Beam Search

Beam search is the de facto search method used for graph-based ANN [43, 59]. We start with a key observation: beam search can be reframed by decoupling the method into two key components 1) a *search order*, determined by a method for traversing the search graph to find candidate nearest neighbors and 2) a *stopping criterion*, which governs when the algorithm stops considering candidates.

Our Adaptive Beam Search method modifies the standard beam search algorithm only by changing the stopping criterion. The search order remains the same. Surprisingly, even this simple change leads to an algorithm that both enjoys strong theoretical approximation guarantees when the underlying graph is navigable (see Theorem 1) and outperforms standard beam search empirically.

The "decoupled view" of beam search is not entirely new. However, for completeness, we detail this reframing in the next section, and show how a change in stopping criterion yields other search algorithms, like simple greedy search and Adaptive Beam Search. We intuitively motivate the stopping criterion used in Adaptive Beam Search before formally analyzing the method in Section 4.

### 3.1 Decoupling Beam Search as Ordered Traversal With a Stopping Condition

To be concrete, we provide pseudocode for a generic version of beam search in Algorithm 1. Implementation details are deferred to Appendix B.1. Importantly, such details do not affect the number of *distance computations* performed by the algorithm – i.e., how many times we evaluate $d(q, i)$ for a query point, $q$, and candidate nearest neighbor, $i$. Distance computations typically dominate the cost of search in practice and, indeed, for the stopping criteria considered in this paper, all other operations can be implemented in time nearly-linear in the number of such computations.

---

**Algorithm 1** Generalized Beam Search

---

**Input:** Search graph $G$ over nodes $\{1, \ldots, n\}$, starting node $s$, distance function $d$, query $q$, target number of nearest neighbors $k$.
**Output:** Set of $k$ nodes $\mathcal{B} \subset \{1, \ldots, n\}$, where each $x \in \mathcal{B}$ is ideally close to $q$ with respect to $d$.

---

1: Initialize min-priority queues $\mathcal{C}$ and $\mathcal{D}$.   ▷ Elements are nodes, priorities are distances to $q$. $\mathcal{D}$ contains all discovered nodes. $\mathcal{C}$ contains discovered nodes that are not yet expanded.
2: Insert $(s, d(q, s))$ into $\mathcal{C}$ and $\mathcal{D}$.
3: **while** $C$ is not empty **do**
4:     $(x, d(q, x)) \leftarrow \text{extractMin}(\mathcal{C})$.                     ▷ Pop min. distance node.
5:     **if** $x$ satisfies **[termination condition]** **then**
6:         **break**
7:     For all $y \in \mathcal{N}_G(x)$, if $y$ is not in $\mathcal{D}$, insert $(y, d(q, y))$ into $\mathcal{C}$ and $\mathcal{D}$.[2]   ▷ Expand node $x$.
8: Obtain $\mathcal{B}$ by running extractMin $k$ times on $\mathcal{D}$, which returns the $k$ elements with the smallest distances from the query, $q$.

---

Algorithm 1 maintains a queue of "discovered nodes" $\mathcal{D}$ whose distances to $q$ have been computed. It repeatedly "expands" the nearest discovered (and not previously expanded) node to $q$ by adding its neighbors to the queue (Line 6). It does so until this nearest node triggers the termination condition in Line 5. The choice of termination condition leads to various versions of greedy search, including beam search and our new distance-based Adaptive Beam Search method. In particular, we have:

**Classic Greedy Search.** Terminate if there are at least:

$$k \text{ items } j_1, \ldots, j_k \in \mathcal{D} \text{ with } d(q, j_i) \leq d(q, x). \tag{1}$$

---

[2]Note that if $\mathcal{D}$ is a simple priority queue, checking if $y \in \mathcal{D}$ may be inefficient. This can be resolved by storing a dictionary of elements in $\mathcal{D}$, as done in our more detailed pseudocode in Appendix B.1.

**Beam Search, with beam-width parameter b ≥ k.** Terminate if there are at least[3] :

$$b \text{ items } j_1, \ldots, j_b \in \mathcal{D} \text{ with } d(q, j_i) \leq d(q, x). \tag{2}$$

**Adaptive Beam Search (our method) w/ parameter $\gamma$.** Terminate if there are at least:

$$k \text{ items } j_1, \ldots, j_k \in \mathcal{D} \text{ with } (1 + \gamma) \cdot d(q, j_i) \leq d(q, x). \tag{3}$$

The rule for greedy search is simple: we terminate if we have already found $k$ points closer to $q$ than the current candidate considered for expansion. For $k = 1$, it takes a moment to confirm that this criterion yields a method that is exactly equivalent to the more typical way of presenting greedy search: starting at $s$, move to the neighboring node nearest to $q$, terminating if there is no neighbor closer than the current node. For $k = 1$, greedy search is known to converge to the exact nearest neighbor if there is some $x \in \{1, \ldots, n\}$ for which $d(x, q) = 0$ *and* the search graph $G$ is navigable [13, 31, 47]. However, no comparable guarantees hold for $k > 1$ or when $q$'s nearest neighbor is not at distance 0, which is typical in practice. Moreover, greedy search performs poorly empirically, easily getting stuck in local minima and failing to find good approximate nearest neighbors.

## 3.2 Relaxing Greedy Search

The goal of beam search is to avoid such accuracy issues. It does so by relaxing the stopping criterion from greedy search: in particular, by (2), we only terminate if we have found $b \geq k$ nodes closer to the query $q$ than our current node $x$. When $b = k$, the algorithms are identical. When $b > k$, greedy search explores a prefix of the nodes explored by beam search, which simply terminates the search at a later point. Beam search is thus guaranteed to obtain a more accurate result than greedy search, at the cost of an increased number of distance computations.

With the above view in mind, many other relaxations of the greedy search termination condition given in (1) become apparent. In (3), we introduce a slack parameter $\gamma \geq 0$ and only terminate if $x$ is further from $q$ than the $k^{\text{th}}$ best discovered point by a factor of $1 + \gamma$. Setting $\gamma = 0$ recovers greedy search, and larger values of $\gamma$ will cause the search process to terminate later, yielding a better result, but at the cost of a higher runtime. This simple idea yields our Adaptive Beam Search procedure.

While intuitively similar to beam search, a key difference of this distance-based termination rule is that it naturally adapts to the *query difficulty*. For simplicity, consider the case of $k = 1$. Greedy search tends to perform worse when there are many "false nearest neighbors". For example, suppose there is just one nearest neighbor $x^*$ with $d(q, x^*) = 1$, but many other points $x_1, \ldots, x_m$ with $d(q, x_i) = 1.01$. Assume that $x_1, \ldots, x_m$, and $x^*$ are all connected with a navigable graph. If the graph is sparse, only a small subset of the nodes in $\{x_1, \ldots, x_m\}$ will be connected to $x^*$. Thus, if we initialize greedy search from a point in $\{x_1, \ldots, x_m\}$, unless we chose a very large beam width $b$, it is likely that before reaching one of the nodes connected to $x^*$ (and thus finding $x^*$), more than $b$ points at distance 1.01 will get added to $\mathcal{D}$, causing the search to terminate. In contrast, as long as $\gamma > .01$, Adaptive Beam Search will continue to search through all of the $x_i$ points, ultimately finding $x^*$ before terminating. At the same time, Adaptive Beam Search will more quickly terminate on easy queries if it becomes apparent that all remaining candidates are too far away to be useful in finding additional nearest neighbors. Indeed, a criterion identical to Adaptive Beam Search has been suggested as an "early stopping" heuristic in work on practical graph-based ANN methods [45, 44].

The intuition that Adaptive Beam Search adapts to query hardness shows clearly in our experiments: as seen in Figure 1, the distribution of distance computations used by Adaptive Beam Search varies more widely, as fewer computations are used for "easier" queries. As a result, across a variety of data sets and search graphs, Adaptive Beam Search consistently outperforms classic beam search in terms of total distance computations required to achieve a certain level of recall for a given query set.

---

[3]*Remark on implementation:* For beam search, it is easy to see that a node $x$ will always satisfy termination condition (2) if it is not one of the closest $b$ neighbors to $q$ in $\mathcal{D}$. So, instead of maintaining two priority queues, it is more computationally efficient to maintain a sorted list of the $b$ closest nodes discovered so far. This is what is done in typical implementations of beam search [59], and in our more detailed pseudocode in Appendix B.1.

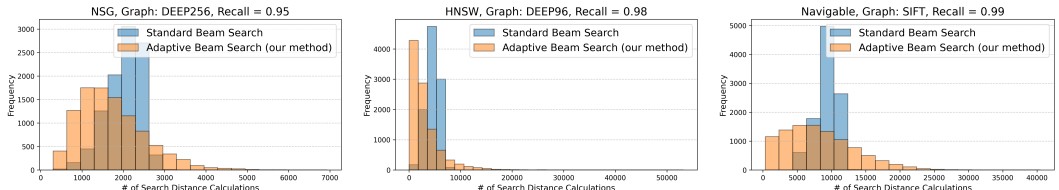

Figure 1: Histograms for the number of distance computations performed by standard beam search and our Adaptive Beam Search method when answering 10,000 queries for various datasets and search graphs (see Section 5 for details). For a fair comparison, the $b$ parameter in beam search and $\gamma$ parameter in Adaptive Beam Search were tuned to achieve a fixed level of recall for the batch of queries. The histograms for Adaptive Beam Search are consistently flatter, confirming the intuition that it better adapts to query difficulty, leading to fewer distance computations on average.

## 4 Theoretical Analysis

We support the improved empirical performance of Adaptive Beam Search with strong theoretical guarantees. Formally, we prove that the method is *guaranteed* to solve the approximate nearest neighbor search problem, assuming that the search graph $G$ is navigable (Definition 1):

**Theorem 1.** *Suppose $d$ is a metric on $\mathcal{X}$ and $G$ is navigable under $d$. Then for any query $q \in \mathcal{X}$, if Adaptive Beam Search – i.e., Algorithm 1 with stopping criterion* (3) *– is run with parameter $0 < \gamma \le 2$, it is guaranteed to return a set of $k$ points $\mathcal{B}$ such that:*

$$\text{for all } v \in \{1, \ldots, n\} \setminus \mathcal{B}, \qquad d(q, v) \ge \frac{\gamma}{2} \max_{j \in \mathcal{B}} d(q, j).$$

Notably, setting $\gamma = 2$, we ensure that all points not returned by the algorithm are at least as far from $q$ as *every point* in $\mathcal{B}$. Thus, **for $\gamma = 2$, Adaptive Beam Search on a navigable graph is guaranteed to *exactly* solve the $k$-nearest neighbor problem.** For smaller $\gamma$, the method obtains an approximate solution: no point in $\mathcal{B}$ can be further from $q$ than any point not returned by more than a $2/\gamma$ factor.[4]

We can see that Theorem 1 proves a trade-off between runtime and accuracy: smaller values of $\gamma$ lead to a strictly faster algorithm (since termination is earlier) but a worse approximation guarantee. While our result falls short of proving worst-case runtime guarantees, to the best of our knowledge, it is the first result linking the accuracy of a natural greedy search method to the notion of graph navigability. Importantly, we note that, unlike our Adaptive Beam Search, a result like Theorem 1 *cannot be proven* for standard beam search. In particular, in Appendix A.2 we prove:

**Claim 2.** *Standard beam search with beam width $b \le n - 3$ fails to approximately solve the nearest neighbor search problem on navigable graphs for any finite approximation factor.*

Concretely, for any finite $C$, we can construct a set of $n$ points in 2-dimensional Euclidean space and a navigable graph $G$ such that, for some query point $q$, beam search run on $G$ with beam width $b \le n - 3$ returns $\tilde{x}$ with $d(q, \tilde{x}) \ge C \cdot \min_{x \in \{1, \ldots, n\}} d(q, x)$.

*Proof of Theorem 1.* Our proof will use the terms "discovered" and "expanded" to identify nodes in $\{1, \ldots, n\}$. We consider a node $j$ "discovered" if $j \in \mathcal{D}$ when Algorithm 1 terminates; i.e., we have evaluated the distance between $j$ and $q$. We consider a node $j$ "expanded" if $j$ is discovered and, at some point, was both popped off $\mathcal{C}$ on Line 4 *and* did not cause the termination condition on Line 5 to be triggered. This ensures that all of its out-neighbors are discovered (see Line 7).

Note that all discovered nodes are added to both $\mathcal{D}$ and $\mathcal{C}$. Formally, if the algorithm terminates because the condition is true for some $x_{term}$, then $\mathcal{C} \cup \{x_{term}\}$ is the set of discovered but not yet expanded nodes, so the set of expanded nodes is $\mathcal{D} \setminus (\mathcal{C} \cup \{x_{term}\})$.

---

[4]Many existing theoretical guarantees for approximate nearest neighbor search, such as those for LSH and related methods [2, 4, 23] focus on the case of $k = 1$. A rephrasing of our result in this case is that Adaptive Beam Search returns an approximate nearest neighbor $\tilde{x}$ with $d(q, \tilde{x}) \le \frac{2}{\gamma} \cdot \min_{x \in \{1, \ldots, n\}} d(q, x)$.

Let $\mathcal{B}$ be the set of nodes returned upon termination and let $\tilde{x} = \text{argmax}_{x \in \mathcal{B}} d(q, x)$ be the $k^{\text{th}}$ furthest point from $q$ in that set. Since $G$ is navigable, and since we assume data points are unique, there must be a path in $G$ from any node $x$ to any other node $y$ (consisting of nodes that get monotonically closer to $y$); i.e., $G$ is strongly connected. Thus, if Algorithm 1 terminates because an empty queue $\mathcal{C}$ causes the while loop to terminate, then all nodes in the graph must have been discovered, and so $\mathcal{B}$ contains the exact $k$ nearest neighbors to $q$, and the theorem holds immediately.

Thus, it suffices to consider the case when termination occurs because some node $x_{term}$ causes the termination condition in Line 5 to evaluate to true and the while loop to break early. We first claim:

**Claim 3.** *When Algorithm 1 terminates, $\tilde{x}$ is guaranteed to have been expanded.*

To see that this claim holds note that, by termination condition (3), it must be that $d(q, x_{term}) \geq (1 + \gamma)d(q, \tilde{x})$ and thus $d(q, x_{term}) > d(q, \tilde{x})$.[5] I.e., $\tilde{x}$ is closer to $q$ then $x_{term}$. Thus, $\tilde{x}$ must have already been popped off $\mathcal{C}$ and expanded before $x_{term}$ was popped off $\mathcal{C}$.

With Claim 3 in place, we can get into our main proof. Our goal is to prove that for all $z \notin \mathcal{B}$,

$$d(q, z) \geq \frac{\gamma}{2} d(q, \tilde{x}). \tag{4}$$

It suffices to prove the claim for all undiscovered nodes $z \notin \mathcal{D}$, since if $z \in \mathcal{D}$ and $d(q, z) < \frac{\gamma}{2} d(q, \tilde{x})$, then $z$ is closer to $q$ than $\tilde{x}$ and would have clearly been included in $\mathcal{B}$ (recall that $\gamma \leq 2$).

Now, suppose by way of contradiction that (4) is not true, i.e., that there is some undiscovered node $z \notin \mathcal{D}$ with $d(q, z) < \frac{\gamma}{2} d(q, \tilde{x})$. We first observe that such a $z$ cannot be an out neighbor of $\tilde{x}$: since $\tilde{x}$ is expanded by Claim 3, all of its neighbors are discovered, i.e., all are in $\mathcal{D}$.

Since $G$ is navigable and all database items are unique, there must be some directed path $\mathcal{P}$ from $\tilde{x}$ to $z$ consisting of points that get monotonically closer to $z$. Moreover, since $z \notin \mathcal{N}_G(\tilde{x})$, $\mathcal{P}$ must have length $\ell \geq 2$. Denote the elements of $\mathcal{P}$ by $\mathcal{P} = \{\tilde{x} = p_0 \to p_1 \to \ldots \to p_\ell = z\}$. We have for all $1 \leq i \leq \ell, d(z, p_{i-1}) > d(z, p_i)$. We make the following claim:

**Claim 4.** *For any $z \notin \mathcal{D}$, there exists some node $w \in \{p_1, \ldots, p_{\ell-1}\}$ along the path from $\tilde{x}$ to $z$ that has been discovered but not expanded.*

*Proof.* First observe that $p_1$ must be discovered since, by Claim 3, $\tilde{x}$ was expanded and $p_1$ is an out-neighbor of $\tilde{x}$. Furthermore, if $p_{i-1}$ is discovered *and* expanded then $p_i$ must be discovered. So, inductively we see that there are two possible cases: either there is some $i < \ell$ for which $p_i$ is discovered but not expanded (as desired) or $p_i$ is discovered *and* expanded for all $i < \ell$. However, the second case is impossible since $z$ is not in $\mathcal{D}$ and it would be if $p_{\ell-1}$ was expanded. We conclude the claim that there is some $w \in \{p_1, \ldots, p_{\ell-1}\}$ that is discovered but not expanded. $\square$

Consider the unexpanded node $w$ guaranteed to exist by Claim 4. When the algorithm terminates,

$$d(q, w) \geq (1 + \gamma)d(q, \tilde{x}). \tag{5}$$

If $w = x_{term}$ this is trivially true as a consequence of the termination rule (3). Otherwise, if (5) were not true, then $w$ would be closer to $q$ than $x_{term}$ and it would have been popped off $\mathcal{C}$ before $x_{term}$ and expanded. With (5) in place, we are ready to obtain our contradiction. By triangle inequality (since $d$ is a metric) and our supposition that $d(q, z) < \frac{\gamma}{2} d(q, \tilde{x})$, we have:

$$d(\tilde{x}, z) \leq d(\tilde{x}, q) + d(q, z) < \left(1 + \frac{\gamma}{2}\right) d(q, \tilde{x}).$$

Combined with another application of triangle inequality and the fact the $d(w, z) < d(\tilde{x}, z)$, we have

$$d(w, q) \leq d(w, z) + d(z, q) < d(\tilde{x}, z) + d(z, q) < \left(1 + \frac{\gamma}{2}\right) d(q, \tilde{x}) + \frac{\gamma}{2} d(q, \tilde{x}) = (1 + \gamma)d(q, \tilde{x}).$$

However, this claim contradicts (5). Thus, there cannot exist any $z \notin \mathcal{D}$ with $d(q, z) < \frac{\gamma}{2} d(q, \tilde{x})$. I.e., (4) holds, proving Theorem 1. For a geometric illustration of the above proof, see Figure 2. $\square$

---

[5] The strict inequality clearly holds when $d(q, \tilde{x}) > 0$ since $\gamma > 0$. When $d(q, \tilde{x}) = 0$ it holds because database items are assumed to be unique, so we cannot also have $d(q, x_{term}) = 0$.

Figure 2: Visualization of the proof of Theorem 1. We let $\tilde{d}$ denote $d(q, \tilde{x})$. Our goal is to show that there is no undiscovered $z$ in a ball of radius $\frac{\gamma}{2}\tilde{d}$ around $q$, which is shown with a dotted line. If there was, we obtain a contradiction. In particular, if $G$ is navigable, we argue that there must be some unexpanded node $w$ on a path of decreasing distance from $\tilde{x}$ to $z$. Since $w$ is closer to $z$ than $\tilde{x}$, it must lie in a ball of radius $\left(1 + \frac{\gamma}{2}\right)\tilde{d}$ around $z$, which is contained in a ball of radius $(1 + \gamma)$ around $q$. However, by (5), no unexpanded node can lie in that ball.

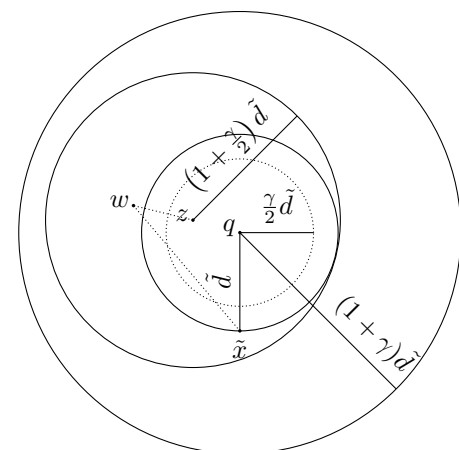

## 5 Experiments

We now experimentally compare our Adaptive Beam Search method with standard beam search, demonstrating improved tradeoffs between efficiency and accuracy in a variety of settings.

**Beam Search Algorithms.** We primarily compare standard beam search (termination condition (2)) with Adaptive Beam Search (termination condition (3)). To implement Algorithm 1 with these termination conditions, we follow the pseudocode in Appendix B.1. For some settings, we test a third method called *Adaptive Beam Search V2*, which terminates on node $x$ if

$$d(q, x) \geq d_1 + \gamma \cdot d_k, \tag{6}$$

where $d_1$ and $d_k$ are the distances from the query $q$ to the closest and $k^{\text{th}}$ closest discovered nodes, respectively. Compared to (3), (6) replaces the threshold $(1 + \gamma) \cdot d_k$ with the smaller threshold $d_1 + \gamma \cdot d_k$, leading to more aggressive stopping. Surprisingly, while (6) is not a relaxation of greedy search (when $\gamma < 1$, it may stop earlier than greedy search), one can check that Theorem 1 still holds under this condition. This motivates its inclusion in our experiments. However, we observe that Adaptive Beam Search V2 generally underperforms Adaptive Beam Search. We leave open developing other stopping conditions that satisfy bounds similar to Theorem 1 while obtaining strong empirical performance like Adaptive Beam Search – see Appendix C.4 for some initial explorations.

**Comparison Across Recall Values.** The algorithms discussed above can all trade off accuracy for runtime by adjusting the beam width, $b$, or the parameter $\gamma$. We thus vary these parameters to obtain a range of *recall values*, i.e., the average fraction of the $k$ nearest neighbors found over all queries on a given dataset. Recall is a standard metric for evaluating ANN methods [43, 59]. We compare the methods by plotting the average number of distance computations performed per query to achieve a certain recall value. Since all three methods have essentially identical implementations, running time scales very similarly with the number of distance computations. See Appendix B.1 for more details.

**Datasets and Graph Constructions.** We evaluate our Adaptive Beam Search on six standard benchmark datasets for nearest neighbor search, which are listed in Table 1. All datasets consist of real-valued vectors in varying dimensions, and we use Euclidean distance for search. We perform evaluations using a variety of popular heuristic "approximately navigable" graphs, along with truly navigable graphs for which the bound of Theorem 1 holds. Specifically, for the heuristic graphs, we use four standard methods: HNSW [43], Vamana [59], NSG [16], and EFANNA [15]. Details on how parameters are set for these algorithms are in Appendix B.3.

### 5.1 Experimental Setup

To construct the truly navigable graphs, we use the approach of [13] to create an initial navigable graph with average degree $\tilde{O}(\sqrt{n})$, and then further prune this graph while maintaining navigability. See Appendix B.2 for details. Pruning reduces the memory footprint of the graph, and results in levels of sparsity closer to those of the heuristic constructions. However, since it is computationally expensive, we only run our navigable graph experiments for random subsets of three of the datasets,

| Dataset | Dimensions | # of Nodes | # of Nodes in Navigable Graph Experiments | # Query Points |
|---|---|---|---|---|
| MNIST [36] | 784 | 60K | 50K | 10K |
| SIFT1M [26] | 128 | 1M | 100K | 10K |
| DEEP96 [7] | 96 | 1M | 100K | 10K |
| DEEP256 [39] | 256 | 1M | - | 10K |
| GloVe [53] | 200 | 1M | - | 10K |
| GIST [50] | 960 | 1M | - | 10K |

Table 1: Datasets used for evaluation. For further details, refer to Appendix B.3.

with subsample sizes listed in Table 1. We believe that our subsample sizes are large enough to be representative. However, it would be interesting to improve the running time of constructing very sparse and truly navigable graphs, so that such graphs can be evaluated for larger datasets.

## 5.2 Results

We now discuss our experimental results on both truly navigable graphs and the commonly used heuristic graphs discussed above.

**Results for Navigable Graphs.** Results for navigable graphs are shown in Figure 3 for SIFT, DEEP256, and MNIST for $k = 1$ and 10. Results for $k = 100$ are included in Appendix C.1. The y-axis shows recall, while the x-axis shows the average number of distance calculations per query. Adaptive Beam Search always performs at least on par with classic beam search, and often significantly better, with up to 30-40% decrease in distance computations for a given recall. Adaptive Beam Search V2 performs worse, so is not evaluated in future experiments. The underperformance of Adaptive Beam Search V2 is further explored in Appendix C.3. In a nutshell, when $d_1 \ll d_k$, for small $\gamma$ we might stop when $d(q, x) < d_k$, which means we do not even explore all the neighbors of our current top-$k$ results. If we increase $\gamma$ to avoid this, we terminate too late when $d_1$ is close to $d_k$.

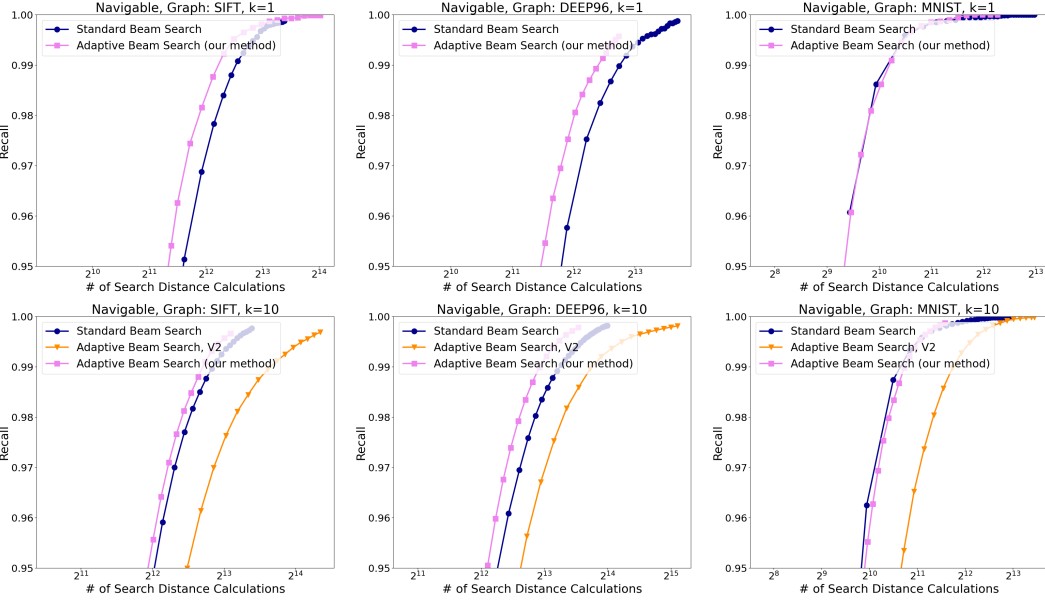

Figure 3: **Navigable Graphs:** Comparison of generalized beam search termination conditions on navigable graphs across three datasets: SIFT1M, DEEP96, and MNIST (columns), with $k = 1$, and $k = 10$ (rows). Adaptive Beam Search consistently outperforms standard beam search, while the alternative Adaptive Beam Search V2 underperforms both by a significant margin. Note that for $k = 1$, Adaptive Beam Search and Adaptive Beam Search V2 are identical, so only one line is shown.

**Results for Heuristic Graphs.** Our results for heuristic graphs with $k = 10$ across three datasets are shown in Figure 4. For additional results covering the remaining datasets and values of $k$, see Appendix C.2. In all cases, we see that Adaptive Beam Search outperforms standard beam search, sometimes marginally, but sometimes by more than a factor of 2, e.g., on MNIST. The performance gains are robust to changing the graph construction, indicating that Adaptive Beam Search is a strong candidate for a drop-in replacement for standard beam search in graph-based ANN.

**Adaptivity Across Queries.** As discussed in Section 3.2, Adaptive Beam Search seems to outperform standard greedy search because the distance-based stopping criterion is more "adaptive" to query difficulty. For hard queries with many approximate nearest neighbors, it tends to use more distance computations. However, the method terminates quickly on easy queries when there are few points with $d(q, x) \leq (1 + \gamma)d_k$. This phenomenon is illustrated for a sample of settings in Figure 1.

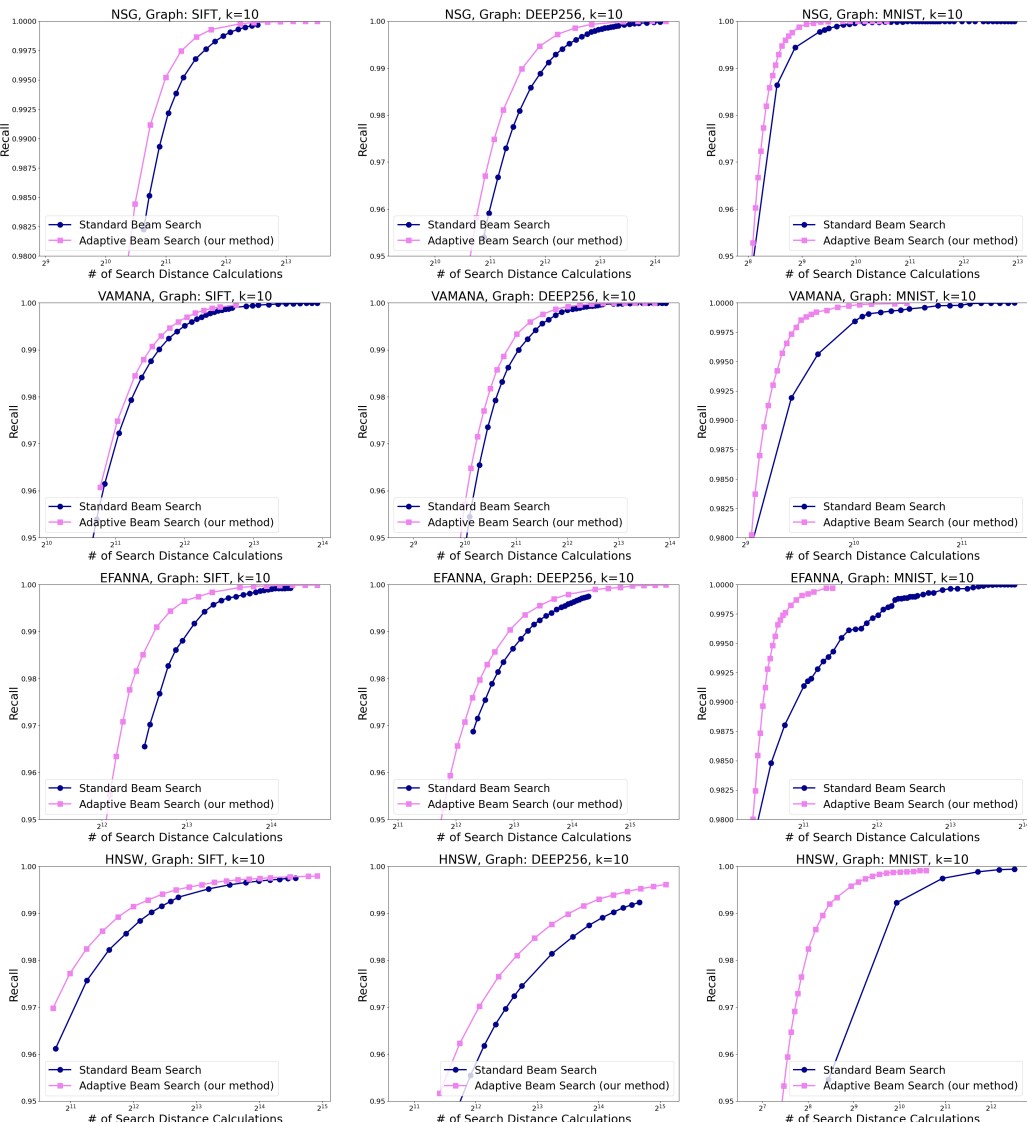

Figure 4: **Heuristic Graphs:** Comparison of generalized beam search termination methods on heuristic graphs produced by NSG, Vamana, EFANNA, and HNSW (rows), for $k = 10$ with 3 datasets: SIFT1M, DEEP256, and MNIST (columns). Adaptive beam search consistently outperforms standard beam search across all cases, sometimes by a significant margin.

## Acknowledgments

We would like to thank Ramon Li for contributions at an early stage of this work. Christopher Musco was partially supported by NSF Award 2106888.

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

# A Additional Proofs

## A.1 Nonexistence of Sparse $\alpha$-Shortcut Reachable Graphs

Recent work of Indyk and Xu [24] shows that, for $k = 1$, standard greedy search (i.e., beam search with beam width $b = 1$) provably returns a $\left(\frac{\alpha+1}{\alpha-1} + \epsilon\right)$-approximate nearest neighbor for any constant $\epsilon$ when run on an $\alpha$-*shortcut reachable search graph* $G$. The $\alpha$-shortcut reachability property requires that, for any nodes $x, y \in \{1, \ldots, n\}$ with $d(x, y) > 0$, there is some $z \in \mathcal{N}_G(x)$ with $\alpha \cdot d(z, y) < d(x, y)$ for some parameter $\alpha \geq 1$. The requirement exactly corresponds to navigability (Definition 1) when $\alpha = 1$ and is a strictly stronger condition when $\alpha > 1$.

The guarantee of [24] is non-vacuous when $\alpha > 1$. Unfortunately, it is also not hard to see that for any fixed $\alpha > 1$, there exist relatively low-dimensional point sets with no sparse $\alpha$-shortcut reachable graphs. In fact, for any constant $\alpha > 1$, it suffices to consider a random point set in $O(\log n)$ dimensional Euclidean space. This contrasts the situation for navigability ($\alpha = 1$), since [11] shows that an $O(\sqrt{n})$ average degree navigable graph can be efficiently constructed for any point set in any dimension (indeed, in any metric space), under the mild assumption of unique pairwise distances between points (which can be ensured, e.g., by tie-breaking with node id). Formally:

**Claim 5.** *For any $\alpha > 1$, let $m = O\left(\frac{\log n}{(1-1/\alpha)^2}\right)$. There are $n$ points in $m$-dimensional Euclidean space with unique pairwise distances, but the only $\alpha$-shortcut reachable graph for the points is the complete graph. Further, by [11], the points admit a navigable graph with $O(\sqrt{n})$ average degree.*

Note that for constant $\alpha > 1$, $1 - 1/\alpha$ is a constant bounded away from 0, so $m = O(\log n)$.

*Proof.* It suffices to find a set of $n$ points whose pairwise distances all lie in the range $(1/\alpha, 1]$. Then, for any $x \neq y$, the only $z$ with $\alpha \cdot d(z, y) < d(x, y)$ is $z = y$. Thus, to ensure $\alpha$-shortcut reachability, all nodes must be connected to all other nodes – i.e., $G$ must be the complete graph.

If we are not concerned about the dimensionality, finding a set of points in Euclidean space with all pairwise distances lying in $(1/\alpha, 1]$ is trivial: take the $n$ standard basis vectors in $\mathbb{R}^n$, scaled by $1/\sqrt{2}$ so that they all have distance 1 from each other. Subtract an infinitesimally small random amount from the non-zero entry of each so that all pairwise distances are unique, but still lie in $(1/\alpha, 1]$.

To obtain a result in lower dimensions, we instead consider random points. Concretely, consider $n$ points in $\mathbb{R}^m$ with each entry set independently to 1 or $-1$ with probability $1/2$. For each $x, y$, we have $\mathbb{E}[\|x - y\|_2^2] = 2m$ and by a standard binomial concentration bounds, $\Pr[|\|x - y\|_2^2 - 2m| \geq m(1 - 1/\alpha)] \leq \exp(-\Omega((1 - 1/\alpha)^2 \cdot m))$. Setting $m = O\left(\frac{\log n}{(1-1/\alpha)^2}\right)$, this probability is bounded by $1/n^c$ for a large constant $c$. Taking a union bound over all $\binom{n}{2} < n^2$ pairs of points, we see that all their squared pairwise distances lie in the range $\left(2m(1 - \frac{1-1/\alpha}{2}), 2m(1 + \frac{1-1/\alpha}{2})\right)$ with probability at least $1 - 1/n^{c-2}$. Normalizing by $2m(1 + \frac{1-1/\alpha}{2})$, all the squared pairwise distances are less than one 1 and greater than $\frac{1 - \frac{1-1/\alpha}{2}}{1 + \frac{1-1/\alpha}{2}} \geq 1 - (1 - 1/\alpha) = 1/\alpha$, where we use the fact that $\frac{1-x}{1+x} \geq 1 - 2x$ for all $x$. Thus, all squared pairwise distances, and in turn all pairwise distances, lie in the range $(1/\alpha, 1)$, as desired. We can again ensure unique pairwise distances by adding arbitrarily small random perturbations to each point, completing the claim. $\square$

## A.2 Failure of Beam Search on Navigable Graphs

We next give a simple counterexample, showing that, unless the beam width is set to essentially the full dataset size, standard beam search on a navigable graph can fail to find an approximate nearest neighbor when run on a navigable graph. This observation in part motivates the definition of our alternative distance-based stopping rule, (3), and the resulting Adaptive Beam Search algorithm.

**Claim 6.** *For any finite $C$, there exists a set of $n$ points in 2-dimensional Euclidean space and a navigable graph $G$ such that, for some query point $q$, beam search run on $G$ with beam width $b \leq n - 3$ returns $\tilde{x}$ with $d(q, \tilde{x}) \geq C \cdot d(q, x^*)$.*

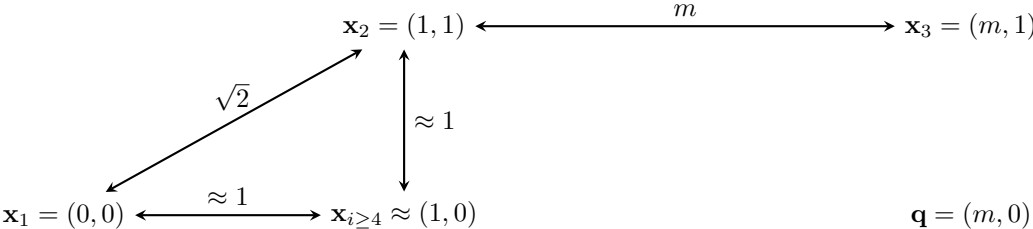

Figure 5: Example showing that standard beam search fails to find a nearest neighbor in a navigable graph. Points $\mathbf{x}_4, \ldots, \mathbf{x}_n$ are all located arbitrarily close to $(1, 0)$. They are all nnected to $\mathbf{x}_1$ and $\mathbf{x}_2$, as well as to each other. The graph is navigable, since we can navigate from $\mathbf{x}_1, \mathbf{x}_4, \ldots, \mathbf{x}_n$ to $\mathbf{x}_3$ and vice-versa through $\mathbf{x}_2$. All other nodes are directly connected to each other. Suppose beam search with beam width $b \leq n - 3$ is initialized at $\mathbf{x}_1$ with query $\mathbf{q}$. Because $\mathbf{x}_4, \ldots, \mathbf{x}_n$ are all closer to the $\mathbf{q}$ than $\mathbf{x}_2$, the method will never expand $\mathbf{x}_2$ and thus fail to reach the nearest neighbor $\mathbf{x}_3$.

*Proof.* Consider the following dataset in 2-dimensional Euclidean space shown in Figure 5: $\mathbf{x}_1 = (0, 0), \mathbf{x}_2 = (1, 1), \mathbf{x}_3 = (m, 1)$ for some arbitrarily large value $m$. Let $\mathbf{x}_4, \ldots, \mathbf{x}_n$ all be located at arbitrary positions in an $\epsilon$ ball around $(1, 0)$ for arbitrarily small $\epsilon$. We can check that the graph with the following two-way edges is navigable: $(\mathbf{x}_2, \mathbf{x}_3)$ and $(\mathbf{x}_i, \mathbf{x}_j)$ for all $i \in \{1, 2\}, j \in \{4, \ldots, n\}$.

Consider beam search initialized at starting point $\mathbf{x}_1 = (0, 0)$ with query $\mathbf{q} = (m, 0)$. The nearest neighbor to $\mathbf{q}$ is $\mathbf{x}_3$ with $\|\mathbf{q} - \mathbf{x}_3\|_2 = 1$. In the first step of beam search, all neighbors of $\mathbf{x}_1$ $(\mathbf{x}_2, \mathbf{x}_4, \ldots, \mathbf{x}_n)$ will be added to the search queue. Since $\mathbf{x}_2$ is further from $\mathbf{q}$ than all nodes in $\mathbf{x}_4, \ldots, \mathbf{x}_n$, the algorithm will then expand nodes from this set in succession, adding no new nodes to the queue since none of these nodes are connected to $\mathbf{x}_3$, the only remaining unexplored node. If $b \leq n - 3$, the algorithm will then terminate, with $\mathbf{x}_2$ never expanded and $\mathbf{x}_3$ never explored.

As a result, beam search returns some $\tilde{\mathbf{x}} \in \{\mathbf{x}_4, \ldots, \mathbf{x}_n\}$ with distance $\|\mathbf{q} - \tilde{\mathbf{x}}\|_2 \geq m - \epsilon$. It thus achieves approximation factor $\frac{\|\mathbf{q} - \tilde{\mathbf{x}}\|_2}{\|\mathbf{q} - \mathbf{x}_3\|_2} \geq \frac{m - \epsilon}{1}$. Setting $m = C + \epsilon$ gives the result. $\qquad\square$

## B  Additional Implementation Details

### B.1  Pseudocode for Generalized Beam Search Variants

Below, we provide detailed pseudocode for generalized beam search (Algorithm 1) under stopping conditions (1) (classic greedy search), (2) (classic beam search), and (3) (Adaptive Beam Search). While the greedy search order and stopping rule determine the number of distance computations performed, it is possible to optimize runtime and storage requirements by using appropriate data structures to implement the stopping rule. Additionally, we can avoid adding nodes to the candidate set $\mathcal{C}$ if we are sure that, if popped off $\mathcal{C}$, those nodes would trigger the termination condition anyways.

**Adaptive Beam Search and Greedy Search.** Pseudocode for Adaptive Beam Search is given in Algorithm 2. The same pseudocode can be used for greedy search, by setting the approximation parameter $\gamma = 0$, so that the Adaptive Beam Search stopping rule (3) becomes the greedy rule (1).

The key optimization is that we maintain a heap, $\mathcal{B}$, of the $k$ nearest points seen so far, which avoids having to extract these neighbors from the set of discovered nodes $\mathcal{D}$ every time termination condition (3) is checked. Further, if a newly discovered node has distance larger than $(1 + \gamma)$ times the $k^{\text{th}}$ closest seen so far, it will always trigger termination if considered for expansion. Thus, we can avoid adding it to the candidate set of unexpanded nodes, $\mathcal{C}$. See Lines 12-17. This optimization avoids letting $\mathcal{C}$ grow unnecessarily large with nodes that will never be expanded.

**Classic Beam Search.** Pseudocode for classic beam search is given in Algorithm 3. The implementation is essentially identical to that of Adaptive Beam Search, except that a heap of the $b \geq k$ nearest points seen so far must be maintained to efficiently check stopping condition (2) each time a node is considered for expansion or newly discovered. At the end of the algorithm, the $k$ nearest points from this heap are ultimately returned. See Lines 22-23.

---

**Algorithm 2** Adaptive Beam Search

---

**Input:** Search graph $G$ over nodes $\{1, \ldots, n\}$, starting node $s$, distance function $d$, query $q$, target number of nearest neighbors $k$, approximation parameter $\gamma$.

**Output:** A set of $k$ nodes $\mathcal{B} \subset \{1, \ldots, n\}$. Each $x \in \mathcal{B}$ is ideally close to $q$ with respect to the distance function $d$.

---

1: $\mathcal{D} \leftarrow \{s\}$                                                            ▷ Dictionary of Discovered nodes
2: $\mathcal{C} \leftarrow \{(s, d(q, s))\}$                                            ▷ Min-heap of candidates
3: $\mathcal{B} \leftarrow \{(s, d(q, s))\}$                                          ▷ Max-heap of best results
4: **while** $\mathcal{C}$ is not empty **do**
5:      $(x, d(q, x)) \leftarrow$ heappop($\mathcal{C}$)
6:      **if** $|\mathcal{B}| = k$ and $(1 + \gamma) \cdot$ findmax($\mathcal{B}$) $\leq d(q, x)$ **then**
7:          **break**                                   ▷ Termination condition from Eq. (3)
8:      **for** all $y \in \mathcal{N}_G(x)$ **do**
9:          **if** $y \notin \mathcal{D}$ **then**
10:             $\mathcal{D} \leftarrow$ insert($\mathcal{D}, y$)
11:             **if** $|\mathcal{B}| < k$ or $d(q, y) < (1 + \gamma) \cdot$ findmax($\mathcal{B}$) **then**
12:                heappush($\mathcal{B}, (y, d(q, y))$)
13:                heappush($\mathcal{C}, (y, d(q, y))$)
14:                **if** $|\mathcal{B}| = k + 1$ **then**
15:                    heappop($\mathcal{B}$)
16: **return** $\mathcal{B}$

---

---

**Algorithm 3** Classic Beam Search

---

**Input:** Search graph $G$ over nodes $\{1, \ldots, n\}$, starting node $s$, distance function $d$, query $q$, beam width $b$, target number of nearest-neighbors $k$.

**Output:** A set of $k$ nodes $\mathcal{B} \subset \{1, \ldots, n\}$. Each $x \in \mathcal{B}$ is ideally close to $q$ with respect to the distance function $d$.

---

1: $\mathcal{D} \leftarrow \{s\}$                                                             ▷ Dictionary of discovered nodes
2: $\mathcal{C} \leftarrow \{(s, d(q, s))\}$                                            ▷ Min-heap of candidates
3: $\mathcal{B} \leftarrow \{(s, d(q, s))\}$                                          ▷ Max-heap of best results
4: **while** $\mathcal{C}$ is not empty **do**
5:      $(x, d(q, x)) \leftarrow$ heappop($\mathcal{C}$)
6:      **if** $|\mathcal{B}| = b$ and **findmax**($\mathcal{B}$) $\leq d(q, x)$ **then**
7:          **break**                                   ▷ Termination condition from Eq. (2)
8:      **for** all $y \in \mathcal{N}_G(x)$ **do**
9:          **if** $y \notin \mathcal{D}$ **then**
10:             $\mathcal{D} \leftarrow$ insert($\mathcal{D}, y$)
11:             **if** $|\mathcal{B}| < b$ or $d(q, y) <$ findmax($\mathcal{B}$) **then**
12:                heappush($\mathcal{B}, (y, d(q, y))$)
13:                heappush($\mathcal{C}, (y, d(q, y))$)
14:                **if** $|\mathcal{B}| = b + 1$ **then**
15:                    heappop($\mathcal{B}$)
16: **for** $i = 1 \ldots (b - k)$ **do**
17:      heappop($\mathcal{B}$).                                 ▷ Reduce $\mathcal{B}$ down to the best $k$ results.
18: **return** $\mathcal{B}$

---

## B.2 Sparse Navigable Graph Construction via Pruning

As discussed, in Section 5, we evaluate the performance of our Adaptive Beam Search method on both truly navigable graphs, where it is backed by the theoretical guarantee of Theorem 1, and on heuristic "approximately navigable" graphs constructed using a variety of popular methods.

To construct sparse navigable graphs, we use the construction of [13]. For $m = \lfloor \sqrt{3n \ln n} \rfloor$, each node is connected to its $m$ nearest neighbors along with $\lceil \frac{3n \ln n}{m} \rceil$ uniformly random nodes. As shown in [13], such a graph is navigable with high probability and has average degree $O(\sqrt{n \log n})$.

We further sparsify these graphs, both to facilitate running large-scale experiments and to more accurately reflect performance on graphs with practical levels of sparsity. To do so, we employ a pruning strategy that removes redundant edges from the graph while maintaining navigability. Pseudocode for the pruning method is given in Algorithm 4. It starts with a navigable graph $G$, then iterates over each node $s$ in the graph, only keeping a minimal set of out edges needed to ensure navigability. In particular, for each node $t \in \{1, \ldots, n\} \setminus \{s\}$, by Definition 1, we must ensure that $s$ has an out neighbor $x$ with $d(x, t) < d(s, t)$. The method iterates over each $t$, adding an out neighbor of $s$ to the *keep* set only if it is needed to ensure this condition holds for some $t$ (i.e., if no edges already in *keep* ensure the condition). After checking all $t$, it removes all neighbors of $s$ not in *keep*.

---

**Algorithm 4** Navigable Graph Pruning

**Input:** Navigable graph $G$ on nodes $\{1, \ldots, n\}$, distance function $d$.
**Output:** Subgraph of $G$ that is still navigable over $\{1, \ldots, n\}$ but ideally has many fewer edges.

1: **for** all $s$ in $\{1, \ldots, n\}$ **do**
2:      keep $\leftarrow \{\}$                            ▷ Set of out neighbors that will remain after pruning.
3:      remove $\leftarrow \mathcal{N}_G(s)$             ▷ Set of out neighbors that will be removed after pruning.
4:      **for** all $t$ in $\{1, \ldots, n\} \setminus \{s\}$ **do**
5:          navigable $\leftarrow$ FALSE
6:          **for** all $x$ in keep **do**
7:              **if** $d(x, t) < d(s, t)$ **then**
8:                 navigable $\leftarrow$ TRUE    ▷ Navigability condition satisfied. No need to add an edge.
9:                 **break**
10:        **if** not navigable **then**
11:           **for** all $y$ in remove **do**
12:              **if** $d(y, t) < d(s, t)$ **then**
13:                 keep $\leftarrow$ keep $\cup \{y\}$     ▷ Keeping edge from $s$ to $y$ ensures navigability to $t$
14:                 remove $\leftarrow$ remove $\setminus \{y\}$
15:                 **break**
16:      **for** all $y$ in remove **do**
17:          $G$.remove_edge$(s, y)$
     **return** $G$

---

The pruning strategy of Algorithm 4 can produce navigable graphs that are significantly sparser than those constructed by [13]. See Table 2 for a summary of the average degrees achieved for our tested datasets. Unfortunately, the runtime of our pruning method scales at least quadratically with $n$. This limits our ability to apply the method to the full datasets. An interesting open question is to improve the running time of constructing very sparse and truly navigable graphs.

| Dataset | Dimensions | # Nodes | Average Out Degree Before Pruning | Average Out Degree After Pruning |
|---------|-----------|---------|-----------------------------------|----------------------------------|
| SIFT1M[26] | 128 | 100K | 3682 | 59 |
| DEEP96 [7, 6] | 96 | 100K | 3682 | 77 |
| MNIST [36, 6] | 784 | 50K | 2516 | 45 |

Table 2: Average out degrees of navigable graphs before and after pruning. Note that we run on subsamples of the full datasets from Table 1 due to the high computational cost of pruning.

### B.3 Omitted Details on Experimental Setup

We next give additional details on the datasets and graphs used to evaluate Adaptive Beam Search.

**Datasets.** Table 1 summarizes the six benchmark datasets used in our experiments. The citation for each dataset includes a note listing the URL where we obtained the specific version of the dataset used in our work. The datasets are available under the following licenses: MIT License (MNSIST), CC0 1.0 Universal (SIFT, GIST), and the Open Data Commons Public Domain Dedication and License (GloVe). We were unable to find license information for Deep96 and Deep256. Both are available in the public domain.

For DEEP96, we used a one-million-point pre-sampled dataset from [6], but our 100K points used for the navigable graph experiments were sampled from the original dataset available at `https://github.com/matsui528/deep1b_gt`. For GloVe, we sampled one million nodes from the original dataset. The GIST data only includes 1K query points by default. To generate 10K query points, in order to match the other benchmarks, we sampled additional query points uniformly at random from the so-called *learning* data points, which are included with GIST for hyperparameter tuning. We did not use this set of points for any other purpose or any parameter tuning.

**Graph Parameters.** As discussed in Section 5, we construct heuristic graphs using four common methods: HNSW [43], Vamana [59], NSG [16], and EFANNA [15]. We used our own implementations of HNSW and Vamana. Code for NSG is available under an MIT License at `https://github.com/ZJULearning/nsg` and for EFANNA under a BSD License at `https://github.com/ZJULearning/efanna`.

The heuristic graph construction algorithms employed take as input various hyperparameters. Settings used for these hyperparameters are given in Table 3. For Vamana, we used the same hyperparameters for all datasets, matching those in the original paper [59], which were found to work well for SIFT, DEEP96, and GIST; using the same parameters for the other datasets yielded similarly good results. The hyperparameters for EFANNA [15] and NSG [16] for SIFT and GIST are taken from authors' repository [16]. The same parameters were also used by [65] and [59]. For NSG and EFANNA with DEEP96, we used the optimal values used by [65]. For EFANNA with MNIST, DEEP256, and GloVe, we tested them using the two set of hyperparameters- the ones used for SIFT and GIST- and picked the better performing. We did a similar thing for NSG with MNIST, DEEP256, and GloVe.

| | EFANNA | | | | | HNSW | | NSG | | | | Vamana | | |
|---|---|---|---|---|---|---|---|---|---|---|---|---|---|---|
| Dataset | K | L | iter | S | R | M | efC | nn | R | L | C | L | R | $\alpha$ |
| SIFT1M | 200 | 200 | 10 | 10 | 100 | 14 | 500 | 200 | 50 | 40 | 500 | 125 | 70 | 2 |
| DEEP96 | 200 | 200 | 10 | 10 | 100 | 14 | 500 | 200 | 50 | 40 | 500 | 125 | 70 | 2 |
| DEEP256 | 200 | 200 | 10 | 10 | 100 | 14 | 500 | 200 | 50 | 40 | 500 | 125 | 70 | 2 |
| GloVe | 200 | 200 | 10 | 10 | 100 | 16 | 500 | 400 | 70 | 60 | 500 | 125 | 70 | 2 |
| GIST | 400 | 400 | 12 | 15 | 100 | 24 | 500 | 400 | 70 | 60 | 500 | 125 | 70 | 2 |
| MNIST | 200 | 200 | 10 | 10 | 100 | 14 | 500 | 400 | 50 | 40 | 500 | 125 | 70 | 2 |

Table 3: Experimental Hyperparameters for Different Datasetsa dn Graph Constructions

For HNSW, we used the hyperparameters that [65] found to be optimal for SIFT, DEEP96, GIST, and GloVe. For HNSW on MNIST and DEEP256, we tested with values of M=14,16,24 and used the best performing on the standard beam search. Since the authors found the ideal value of efc for SIFT, DEEP96, GIST, and GloVe to be 500, we used this value for DEEP256 and MNIST.

**Computational Resources.** Navigable graphs were constructed using our pruning methods run on a single core of a 3.2GHz Intel Core i9-12900K CPU with access to 128GB of DDR5 4800mhz RAM. To accelerate pruning and take advantage of available memory, we precomputed all pairwise distances between pairs of points in the dataset. Each graph required several hours to construct. All other experiments were run on a single 2.9GHz Intel(R) Xeon(R) Platinum 8268 CPU with access to 32GM of RAM, although at most 4GB was used for any individual experiment. Producing a single recall/distance computation tradeoff curve requires several hours for each dataset and algorithm.

## C  Additional Experimental Results

In this section, we include additional experimental results.

### C.1  Navigable Graphs

In Figure 6 we compare beam search termination conditions on three datasets for $k = 100$. The results are similar to those reported in Figure 3 for $k = 1$ and $k = 10$, but with less significant gains seen for Adaptive Beam Search as compared to standard beam search. As for smaller values of $k$, Adaptive Beam Search V2 underperforms both other methods.

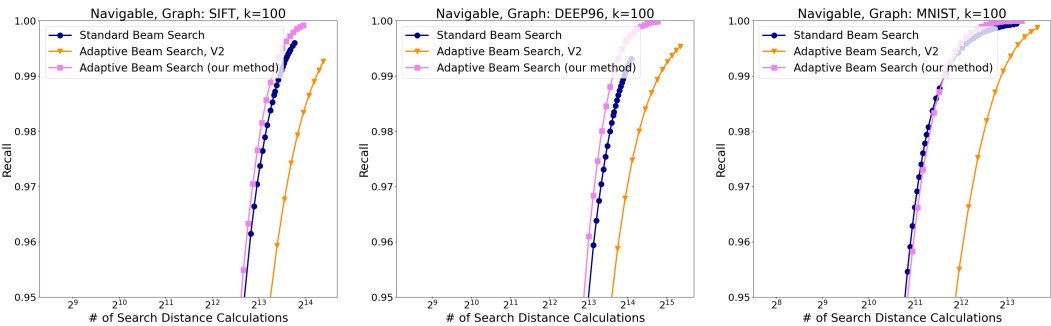

Figure 6: Comparison of generalized beam search termination conditions on navigable graphs across three datasets: SIFT1M, DEEP96, and MNIST (columns), with $k = 100$ (rows). Adaptive Beam Search consistently outperforms standard beam search, while the alternative Adaptive Beam Search V2 underperforms both by a significant margin.

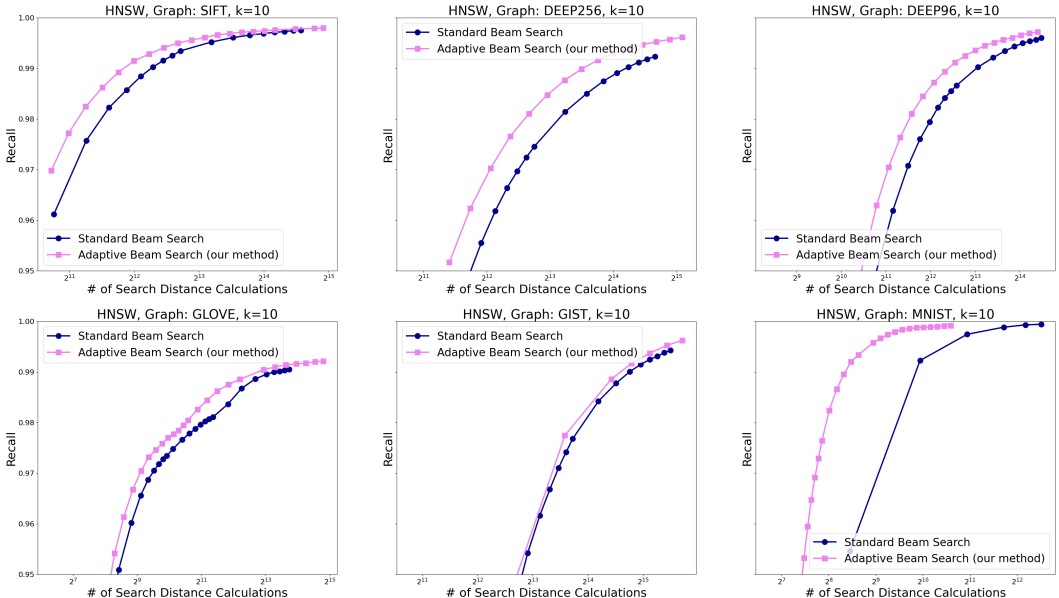

Figure 7: Comparison of generalized beam search termination methods on HNSW graphs with $k = 10$ across six datasets. Adaptive Beam Search outperforms standard beam search, with the degree of improvement varying across datasets.

## C.2   Heuristic Graphs

In Figure 7 we compare beam search termination conditions on HNSW search graphs for all six benchmarks and $k = 10$. In Figure 8, we include further results on HNSW graphs for $k = 1$ and $k = 50$ across three datasets. As with our other experiments on heuristic graphs (see Figure 4), we see that Adaptive Beam Search generally outperforms standard beam search, sometimes by a large margin. One exception is for GIST with $k = 1$, where beam search performs marginally better.

## C.3   Adaptive Beam Search vs. Adaptive Beam Search V2

As illustrated in Figure 3, Adaptive Beam Search V2, which uses the more aggressive stopping condition of (6), generally underperforms both Adaptive Beam Search and classic beam search. We believe this is due to the fact that, to achieve high recall, the $\gamma$ parameter for this rule needs to be set high, causing the method to terminate late and perform a large number of distance computations on some queries. This phenomenon is illustrated in Figure 9.

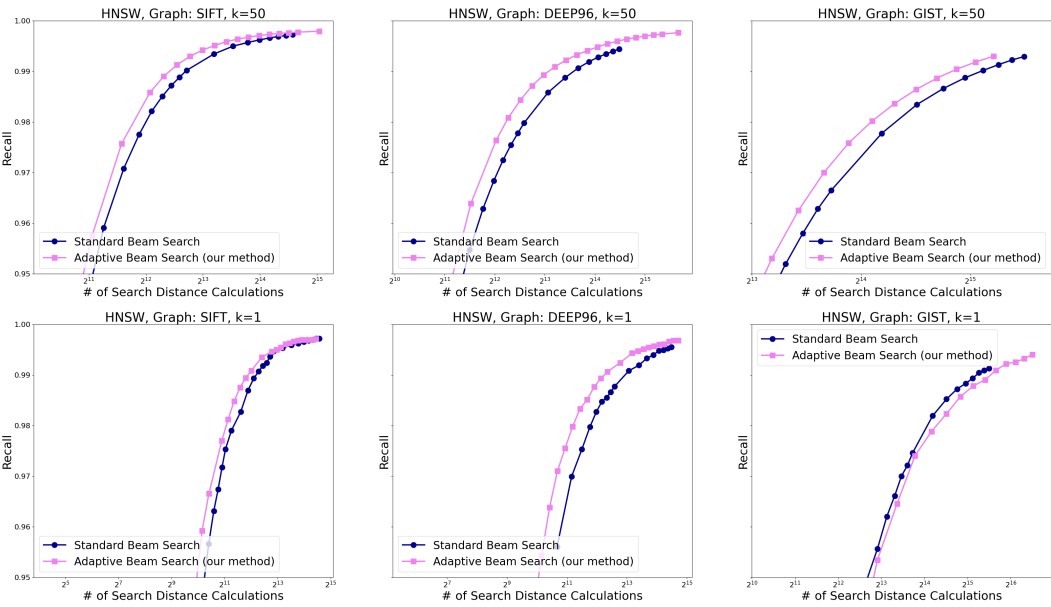

Figure 8: Comparison of generalized beam search termination methods on HNSW graphs across three datasets with $k = 50$ and $k = 1$. Adaptive Beam Search outperforms standard beam search as we vary $k$, with the exception of GIST for $k = 1$, where it slightly underperforms.

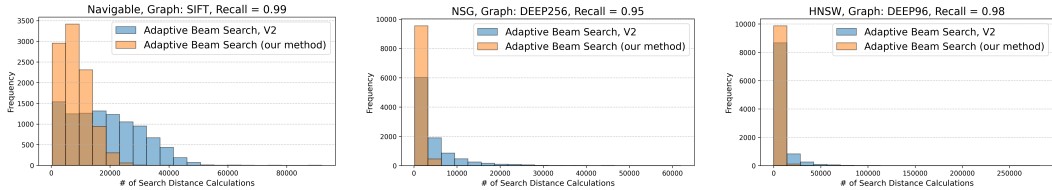

Figure 9: Histograms for the number of distance computations performed by Adaptive Beam Search and Adaptive Beam Search V2. We tune the $\gamma$ parameter for each method to achieve a fixed recall value, finding that Adaptive Beam Search V2 has a heavier tail of queries that require many distance computations, in part explaining its poor performance seen in Figure 3.

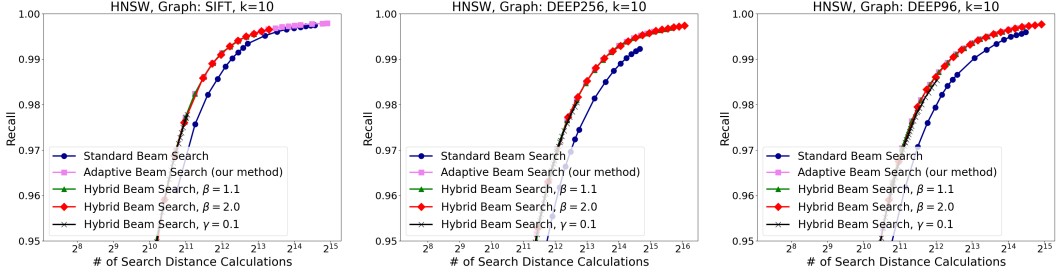

Figure 10: Evaluation of the Hybrid Beam Search termination rule from (7) on three datasets. There is very little difference in performance between the method and Adaptive Beam Search.

## C.4 Hybrid Stopping Rule

As discussed in Section 5, it would be interesting to consider other relaxations of greedy search beyond beam search and Adaptive Beam Search. For example, [21] considers a rule similar to *Adaptive Beam Search V2* (6): instead of using the stopping criteria $d(q, x) \geq (1 + \gamma) \cdot d_k$ as in Adaptive Beam Search, they use criteria $d(q, x) \geq d_k + \gamma \cdot d_1$, where $d_1$ and $d_k$ are the distances

from the query $q$ to the closest and $k^{\text{th}}$ closest discovered nodes, respectively. Initial experiments on this approach suggest that it performs very similarly to Adaptive Beam Search.

Another obvious candidate is a stopping rule that combines the distance-based relaxation of Adaptive Beam Search and the rank-based stopping rule of standard beam search. In particular, in Algorithm 1 we could choose to terminate if there are at least:

$$b \text{ items } j_1, \ldots, j_b \in \mathcal{D} \text{ with } (1 + \gamma) \cdot d(q, j_i) \leq d(q, x), \tag{7}$$

where $b > k$ is a "width parameter" and $\gamma > 0$ is a distance-based relaxation. We ran initial experiments with this natural hybrid termination, which are shown in Figure 10. To obtain a trade-off curve between recall and distance computations, we either fixed $b = \beta \cdot k$ for a parameter $\beta > 1$ and then varied $\gamma$, or we fixed $\gamma$ and varied $\beta$. Somewhat surprisingly, the hybrid method appears to perform very similarly to Adaptive Beam Search, although further study of this termination condition and other relaxations would be valuable.

