# OpenReview forum: "Distance Adaptive Beam Search for Provably Accurate Graph-Based Nearest Neighbor Search"
_NeurIPS.cc/2025/Conference — NeurIPS 2025 poster_

### Official Review · Reviewer_Lky8 · 2025-06-10

**Clarity:** 2
**Significance:** 2
**Originality:** 2
**Rating:** 2
**Confidence:** 4

**Summary:**

The paper introduces a new search algorithm that adjusts the termination condition according to distance and theoretically demonstrates its ability to approximately solve the nearest neighbor problem on the navigable proximity graph index. Extensive experiments on real-world datasets validate the efficacy of the proposed algorithm.

**Questions:**

Please refer to W1-W7.

**Ethical Concerns:**

["NO or VERY MINOR ethics concerns only"]

**Final Justification:**

After the rebuttal, my main concerns are still as follows.

- The provable approximation guarantee is based on the assumption of graph navigability, a condition that is challenging to meet in practical scenarios, thereby limiting the practical utility of these guarantees. Since the navigability graph may have a large number of degrees, line 7 in Algorithm 1 can not achieve good performance when considering the guarantee.
- The performance improvements proposed seem relatively modest, especially for lower values of \(k\) and dimensions. Insufficient experimental evidence, particularly in the absence of further results, raises concerns. The clarification states that on the MNIST dataset, no significant differences were observed across approaches, highlighting the necessity for broader experimentation using datasets like those from ANN and BIGANN benchmarks to substantiate effectiveness.

Considering the aforementioned points, I maintain my negative score.

**Limitations:**

Yes

**Quality:**

2

**Strengths And Weaknesses:**

**Strengths**

S1. The paper studies an important problem.

S2. The paper theoretically verifies the efficacy of the proposed algorithm on navigable graphs.

S3. The algorithm presented in this paper can be seamlessly integrated into all proximity graph approaches through simple modifications.

**Weaknesses**

W1. The definition of beam search on line 159 deviates from existing approaches, leading to confusion regarding their equivalence, necessitating further discussion.
- In traditional approaches, the queue in beam search has a width of $b$, exploring iteratively until all neighbors of nodes in the queue are farther from the query vector.
- In contrast, this paper defines the terminal condition as when the top-$(b+1)$ node in $\mathcal{D}$ is the explored node.
- The key distinction lies in the fact that the queues $\mathcal{C}$ and $\mathcal{D}$ in the search are infinite, whereas the queue size in beam search is limited to $b$. It may remove a node from the queue if it is not currently in the top-$b$, but its neighbor could still be one of the top-$k$. If it remains in $\mathcal{C}$ without removal. Is the situation the same?

W2. The concept of decoupling beam search into two queues has been implemented in HNSW [1] and further proposed in [2]. Therefore, it is essential to discuss and emphasize the differences here; otherwise, the contribution may not be considered original.

W3. The paper [3] presents the beam search on proximity graphs involves two phases. Has the author considered setting different values of $\gamma$ for the two phases to potentially enhance search performance?

W4. The enhancement offered by the proposed search algorithm appears relatively minor, particularly for small values of $k$ and low dimensions. Is the performance correlated with the values of $k$ and $d? More experimental results may be necessary to delve into this aspect.

W5. Transitioning from a navigable graph to a practical version poses challenges due to the absence of navigable paths between all node pairs. Thus, beam search is suggested to enlarge the queue size and overcome local optima. However, the rationale behind adaptive beam search in the practical proximity graph index version is unclear and requires further discussion.

W6. The example presented in line 185 is perplexing. Why does beam search require $m+1$ to find $x^\*$? Is $x^\*$ a neighbor of one of $y_i$?

W7. At line 119, the statement "navigable graphs can in general be much sparser than $\alpha$-shortcut reachable graphs" raises an issue. Releasing the $\alpha$ condition implies that it is easier for $x$ to have a neighbor $z$ such that $\alpha \cdot d(z,y) < d(x,y)$. Hence, $x$ may have fewer neighbors to achieve $\alpha$-shortcut reachable graphs compared to navigable graphs.

[1] Hnswlib - fast approximate nearest neighbor search. https://github.com/nmslib/hnswlib.

[2] Jianyang Gao, Cheng Long. High-Dimensional Approximate Nearest Neighbor Search: with Reliable and Efficient Distance Comparison Operations. SIGMOD 2023.

[3] Zhang et al. VBASE: Unifying Online Vector Similarity Search and Relational Queries via Relaxed Monotonicity. OSDI 2023.

---

> ### Author Rebuttal · Authors · 2025-07-30
>
> Thank you for the feedback. We respond to specific questions below:
>
> **W1:** We can add further discussion of the equivalence. In short, for standard beam search, if at any point a node is in the $(b+1)$st or higher position in $\mathcal{C}$, it will never be “explored”. In particular, this means that at least $b$ other nodes are closer to the query, so the termination condition is guaranteed to trigger immediately if that node is ever popped off $\mathcal{C}$. Accordingly, the neighbors of a node not currently in the top $b$ are never added to $\mathcal{C}$. Moreover, for this reason, the method does not change if the length of $\mathcal{C}$ is truncated to $b$, which is the more standard implementation suggested by the reviewer. We hope this clarifies any confusion.
>
> **W2:** We do not claim splitting into two queues as a contribution. We view this as more of an implementation detail. As for “decoupling” into a search order + termination rule, it does seem that [2] shares this perspective, and we will be sure to add a citation. To be clear, in the paper we wrote: “We suspect the “decoupled view” of beam search is not novel, but we have not seen it presented”. The main novelty in our paper is observing that a simple alternative to the beam search termination criteria yields strong provable approximation guarantees when run on navigable graphs, and these guarantees also translate into better empirical performance.
>
> **W3:** We have not, but it is an interesting direction for future work. We value the simplicity of Adaptive Beam Search (which, like standard beam search, only involves a single hyperparameter). However, multiple phases could indeed be helpful.
>
> **W4:** The x-axis of all plots is on a log scale, so typical improvements range between a 10% and 50% reduction in distance computations. We consider this significant for a problem of such practical importance, especially given that Adaptive Beam Search is a very light-weight, simple to implement modification of Beam Search. That said, we do think there is room for further improvement! We did not notice a strong correlation with the improvement and $k$, although we agree that this is worth exploring more. We did notice that some datasets resulted in very little differences between the methods. MNIST is a good example. We believe this is because most MNIST queries have a very close nearest neighbor in the dataset. For such queries, even basic greedy search should perform quite well (indeed, basic greedy search always succeeds on a navigable graph if the query is exactly in the dataset). This leaves little room for improvement for Adaptive Beam Search or standard beam search for datasets like MNIST, so it makes sense that they perform similarly.
>
> **W5:** It is important to point out that local optima are an issue even in navigable graphs – standard greedy search is not guaranteed to return a good approximate nearest neighbor. Thus, the main goal of Adaptive Beam Search is to escape local minima exactly as the reviewer described. It does so by also “enlarging the queue”, just like standard beam search. The difference is that the amount of enlargement is “adaptive” rather than restricted to a fixed upper limit, b. As illustrated in Figure 1, this allows Adaptive Beam Search to spend more effort on more challenging queries. We can add more discussion about this intuition in the paper.
>
> **W6:** Thanks for the question – we will clarify this example. We agree that the current explanation is incomplete. To be more concrete, have the points $y_1,...,y_m$ all be equidistant from the query point $q$ (with distance $1.01$). Have $y*$ be the closest point in $y_1,...,y_m$ to $x*$. Have $y*$ connected to $x*$ and all $y_1,...,y_m$ points connected to each other. We can check that this graph is navigable. However, suppose we start search at $y_1$. $x*$ will only ever be found if we explore its neighbor $y*$. We won't do so in standard beam search with beam width $< \Omega(m)$ if we assume node ids are arbitrary (since $y*$ will not be preferenced for exploration over any of $y_1$'s $m$ other neighbors).
>
> **W7:** Any $\alpha$-shortcut reachable graph for  any $\alpha \geq 1$ (which is the relevant parameter regime studied in prior work) is also navigable. In particular, the existence of $z$ with $\alpha d(z,y) < d(x,y)$ implies the existence of $z$ with $d(z,y) < d(x,y)$. Thus, it is always possible to construct a navigable graph at least as sparse as any $\alpha$-shortcut reachable graph (take the $\alpha$-shortcut reachable graph itself). In many cases, navigability allows for much higher levels of sparsity. E.g., in Appendix A.2, we discuss random point sets for which any $\alpha$-shortcut reachable graph for fixed $\alpha > 1$ requires $\Omega(n^2)$ edges. At the same time, by citation [12] in the paper, there exist navigable graphs with just $O(n^{3/2})$ edges for these (and all) point sets.

---

> > ### Comment · Reviewer_Lky8 · 2025-08-03
> >
> > Thank you for the author's clarifications.
> >
> > > The main novelty in our paper is observing that a simple alternative to the beam search termination criteria yields strong provable approximation guarantees when run on navigable graphs, and these guarantees also translate into better empirical performance.
> >
> >
> > My main concerns are as follows.
> >
> > - The provable approximation guarantee is based on the assumption of graph navigability, a condition that is challenging to meet in practical scenarios, thereby limiting the practical utility of these guarantees. Since the navigability graph may have a large number of degrees, line 7 in Algorithm 1 can not achieve good performance when considering the guarantee.
> > - The performance improvements proposed seem relatively modest, especially for lower values of \(k\) and dimensions. Insufficient experimental evidence, particularly in the absence of further results, raises concerns. The clarification states that on the MNIST dataset, no significant differences were observed across approaches, highlighting the necessity for broader experimentation using datasets like those from ANN and BIGANN benchmarks to substantiate effectiveness.
> >
> > Considering the aforementioned points, I maintain my negative score.

---

> > > ### Author Response · Authors · 2025-08-05
> > >
> > > To these points we want to emphasize that:
> > >
> > > 1) Some assumption on the search graph needs to be made to prove theoretical guarantees. Our navigability assumption is *strictly weaker* than other properties that have been previously assumed to prove guarantees, like the $\alpha$-shortcut reachability assumption from [Indyk, X,  NeurIPS 2023] and [Gollapudi et al, ICML 2025].
> > >
> > > Moreover, we do not think that the fact that current heuristics fail to return navigable graphs should be taken as evidence that the condition is challenging to achieve. As shown in Table 2, even a simple heuristic constructs provably navigable graphs with degree between 45 and 77 for the datasets we tested on. For comparison, Microsoft's DiskANN documentation recommends choosing a search graph degree between 60 and 150. Of course, there is more to explore here, and graph construction is not the focus of our work, but there is active work on developing fast algorithms for constructing sparse navigable graphs (see e.g. the recent arXiv print "Efficiently Constructing Sparse Navigable Graphs" by Conway et al.
> > >
> > > 2) Our paper contains 33 experiments across 6 datasets, 5 graph constructions, and various $k$ values. In every experiment but one, Adaptive Beam Search outperforms standard beam search along *all points* of the recall curve. For the remaining one experiment ($k=100$, MNIST, navigable graph) the methods are basically identical. While running more experiments is always possible, we believe this is already strong evidence to consider Adaptive Beam Search over Standard Beam Search. Graph based nearest neighbor search is currently deployed at industrial scale, so savings of even 10-50% in compute costs are considerable.
> > >
> > > Best,
> > > The authors

---

> > > > ### Comment · Reviewer_Lky8 · 2025-08-08
> > > >
> > > > Thank you for the clarification. However, my concerns persist:
> > > >
> > > > - The assumption appears weaker than existing works, but it does not imply that it has practical significance.
> > > > - The dataset size for evaluation is limited, not exceeding one million vectors. This size may not adequately demonstrate the scalability of the proposed algorithms. Furthermore, no additional experimental results were presented during the rebuttal phase.
> > > > - Table 2 displays degree results. Are these graphs theoretically navigable, or do they represent a practical version that sacrifices certain properties?
> > > > - I doubt that Adaptive Beam Search will supersede Standard Beam Search. Standard Beam Search offers a parameter to regulate the number of final results, making it practical for real-world applications. In contrast, Adaptive Beam Search lacks this control over final results. In cases where results are insufficient, how can we proceed to obtain more results? If we consider the larger parameters in Adaptive Beam Search, it means restarting the search, which is a waste.

---

> > > > > ### Author Response · Authors · 2025-08-08
> > > > >
> > > > > To answer the question: Yes, those graphs are theoretically navigable. We agree that further work is needed (ideally involving larger datasets) to explore the possibility of working with truly navigable graphs in practice. This is something that we and others in the community are actively working on. Nevertheless, our initial results are promising.

---

> > > > > > ### Comment · Reviewer_Lky8 · 2025-08-08
> > > > > >
> > > > > > However, constructing a theoretically navigable graph is impractical due to the huge construction cost, suggesting that theoretical guarantees may not be useful in real-world applications, which is too ideal. Additionally, due to the incompleteness of the experiments and the lack of comparison to related works, as noted by another reviewer, I do not support acceptance at this moment.

---

### Official Review · Reviewer_gkoF · 2025-06-28

**Clarity:** 4
**Significance:** 3
**Originality:** 3
**Rating:** 5
**Confidence:** 4

**Summary:**

The paper suggests a query algorithm for nearest neighbor search within the graph-based framework for storing and processing large-scale data. Nearest neighbor query algorithms for large, high-dimensional data have become important given that such queries are frequently used for mining vector datasets generated by large language models. One way to store and process such large vector datasets is to construct a sparse directed graph where the nodes are the data items, and the neighbors of a node are cleverly chosen so that a nearest neighbour query q can be processed by moving along the nodes of this search graph in the direction where the distance from q decreases. There are techniques to construct such graphs so as to make query processing fast. Construction of this search graph is not the subject of discussion in this paper. The main subject of discussion is the query algorithm over "navigable graphs." These are graphs such that for every query node q and node x (it'll help to think this as the current node during exploration), if d(q, x) > 0, then there is a neighbor y of x such that d(q, y) < d(q, x).

Suppose we want k nearest neighbors of a query point q. The search algorithm follows this general template (taken from the paper):
- Maintain two priority queues C and D. Initialise C with (s, d(q, s)) for an arbitrary node s.
- While (C is not empty):
	- x = extractMin(C)
	- if (x satisfies 'termination condition'): break
	- For all y in N(x), if y is not in D, insert (y, d(y, q)) into C and D
- Return k nearest neighbors by running extractMin k times on D

This general template allows us to write various search strategies that are used. We just need to specify the 'termination_condition':
1. Greedy search: if there are k nodes in D with a smaller distance from q than x
2. Beam search: If there are b (b > k) nodes in D with a smaller distance than x.

The above two techniques have been used in the past for query search. The paper suggests the following method, which they call adaptive beam search:
3. Adaptive beam search: If there are k nodes v1, ..., vk in D such that (1+\gamma)*d(q, vi) < d(q, x) for every i.

The tradeoff is between speed and accuracy. Beam and Adaptive beam search may improve the accuracy, but at the cost of increasing the number of distance calculations. The paper shows the advantage of using adaptive beam search over beam search in the following aspects:
1. There are much better theoretical guarantees for accuracy on navigable graphs given by adaptive beam search over beam search. For adaptive beam search, it is shown in the paper that for \gamma=1, the query algorithm will output approximately accurate results. On the other hand, it is shown in the paper that for beam search, one can construct an input dataset on which the algorithm performs accurately only at a huge cost of distance calculations.
2. Experimental results show that for the same value of accuracy, adaptive beam search uses a lesser number of queries for most datasets.

**Questions:**

Comments:

- It will be good to also include the greedy search in the experimental results to get a comparison.
- I do not see the point of including adaptive search V2 in the experimental analysis.
- Some other questions are mentioned in the summary and strengths and weaknesses.

**Ethical Concerns:**

["NO or VERY MINOR ethics concerns only"]

**Final Justification:**

I remain positive about the paper after the rebuttal. My concerns have been addressed in the rebuttal. I will maintain my original score.

**Limitations:**

Yes. The paper presents theoretical results that do not have any potential negative societal impact.

**Paper Formatting Concerns:**

There are no serious paper formatting concerns.

**Quality:**

3

**Strengths And Weaknesses:**

Strengths:
- The paper discusses a problem that is current and very relevant in AI-powered systems. The improvement in query time will have a significant impact on many applications.
- The theoretical guarantees are interesting.

Weaknesses:
- The negative result for beam search (which is for query complexity) seems incomparable to the positive result for adaptive search (which is for accuracy). Are there negative results related to query complexity also for adaptive beam search?

---

> ### Author Rebuttal · Authors · 2025-07-30
>
> Thank you for the positive feedback. Answers to specific comments and questions are below:
>
> **Re: Query Complexity Bound.** We think Claim 2 (the beam search lower bound) is comparable to Theorem 1 in the following sense: Theorem 1 shows that we can obtain a provable approximation bound when the Adaptive Beam Search method is run with a fixed relaxation parameter  that only depends on the desired approximation quality. On the other hand, Claim 2 shows that, setting the beam width $b  = c*k$ for essentially any reasonable factor $c > 1$ won’t guarantee that a good approximation is returned from standard beam search. Indeed, if $b = n$, then beam search performs an exhaustive search, and our lower bound holds all the way up to $b = n-3$. The goal was to rule out the existence of a comparable result to Theorem 1 for standard beam search – e.g., we rule out results of the flavor “setting $b  = (1+\gamma)*k$ yields to a $1/\gamma$-approximation.”
>
> That said, it is interesting to ask about both upper and lower bounds on number of distance computations (which we believe is what the reviewer means by query complexity?). It is possible to construct adversarial navigable graphs for which even Adaptive Beam Search requires many distance computations. For example, consider points arranged along a one-dimensional line. If these points are connected via a bidirectional path, the resulting graph is navigable. However, a typical query (even when the query point is in the dataset) will require $\Omega(n)$ distance computations to answer.
>
> Nevertheless, as discussed in our response to Reviewer gJ9t03, citation [12] in the paper shows how to construct sparse navigable graphs with "low depth" – i.e., greedy search terminates in just two steps for queries that are in the dataset. This implies sublinear query complexity on average for query points in the dataset. Thus, there is hope to give query/computational complexity upper bounds for sparse navigable graphs satisfying such a low-depth or related condition. We believe this would be an exciting direction for future work.
>
> **Re: Greedy Search in experiments.** We did not include greedy search in our plots since there is not a natural way to trade-off between distance computations and recall. Indeed, greedy search is a special case of beam search with $b = k$ or Adaptive Beam Search with $\gamma = 0$. In this sense, it produces a single point to the far left of the curves for Adaptive Beam Search and Standard Beam Search, which often has such low recall, it would be off the charts. That said, we think this point is worth emphasizing, so will consider including some annotation for the plots to indicate what specific distance computation/recall point greedy search would have produced.
>
> **Re: Adaptive Search V2.** Another reviewer commented on this as well, and the point is well taken. We will either appendicize or remove entirely.

---

> > ### Comment · Reviewer_gkoF · 2025-08-04
> >
> > I would like to thank the authors for the detailed response. I have gone over the rebuttal carefully. I will maintain my positive score.

---

### Official Review · Reviewer_ZV7M · 2025-06-29

**Clarity:** 3
**Significance:** 2
**Originality:** 2
**Rating:** 3
**Confidence:** 5

**Summary:**

This paper proposes Adaptive Beam Search to acclerate GANNS by replacing the classical beam search in order to reduce the expanded nodes in the graph. It provides theoretical guarantees in navigable graphs.  The authors also conduct experiments on real-world datasets to verify the proposed method.

**Questions:**

Q1. Could you compare your method with the three baselines mentioned above? i.e., (1) GGNN: Graph-Based GPU Nearest Neighbor Search (2) earned adaptive early termination (ref. [33] in this work) and (3) Jiadong Xie et. al. Graph Based K-Nearest Neighbor Search Revisited.

Q2. Please specify the parameter selection in your experiments.

If those questions could be properly addressed, I would raise my rating.

**Ethical Concerns:**

["NO or VERY MINOR ethics concerns only"]

**Limitations:**

Yes

**Quality:**

2

**Strengths And Weaknesses:**

S1. The authors proposed an interesting method for GANNS and established connections between search performance and navigable graphs in theory.

S2. The experimental results are impressive, which reduces distance computations up to 50%.

S3. The method is easy to implement and use in practice, which may be interesting for practical users or engineers.

W1. The theoretical analysis depends on the assumption of navigability of the graphs, while widely-used HNSW, NSG and Vamana are just approximately navigable.

W2. The experiments are not comprehensive. (1) There is no study on the effects of the parameter \gamma. (2) There is no comparisons with the baselines such as learned adaptive early termination (ref. [33] in this work). (3) Another similar work is not mentioned in this work, i.e.,  Fabian Groh et.al.  GGNN: Graph-Based GPU Nearest Neighbor Search. GGNN also modify the termination condition of the beam search via a distance slacking factor. (4) It is not clear how the parameters of graphs are selected, which are key to the search performance.

W3. The authors ignore the recent progress in GANNS, i.e., Jiadong Xie et. al. Graph Based K-Nearest Neighbor Search Revisited, in TODS. This work take an adaptive strategy on online selected neighbors of expanded nodes. This work should also be considered as the baseline.

W4. The technique depth is kind of limited, since it only modifies the termination condition.

---

> ### Author Rebuttal · Authors · 2025-07-30
>
> We thank the reviewer for their helpful feedback and address specific concerns and questions below.
>
> **W1:** This is true, however, given the diversity of graph construction algorithms, any broadly applicable theoretical results for graph-based near neighbor search will necessarily need to make some assumptions about the resulting graph produced. We believe that navigability is a reasonable assumption since 1) it is explicitly discussed as a target/inspiration for these heuristic graph constructions, 2) sparse navigable graphs are known to exist for all datasets [12], and 3) our method gives performance improvements not just on navigable graphs but on these popular heuristic constructions, evidencing that our theoretical results are still informative in practice. Prior theoretical results, such as those of Indyk and Xu, NeurIPS 2023 for graph-based near neighbor search require strictly stronger assumptions on the underlying graph, in particular, $\alpha$-shortcut reachability.
>
> **W2 (1)**: All of the plots in the experiments section that plot recall vs. distance computations explore the effect of varying $\gamma$: we obtain these plots by starting with a small value for $\gamma$ and increasing its value in small increments, which strictly increases both the number of distance computations and the recall. For example, the first plot in Figure 2 uses gamma values ranging between .1 and .65. Nevertheless, we agree with the reviewer that it would be helpful to clarify this point in the writing. We propose to add annotations to the points on a few of the plots showing what choice of gamma led to what point.
>
> **W2 (2) and Q1**:
> We were able to run comparisons with the Fabian Groh et. al paper. Thank you for pointing this paper out – we were unaware of the method. Indeed, that paper uses an additive instead of a multiplicative slack factor. While we cannot share new plots due to the rules for this year’s rebuttal, in our experiments, the Groh et. al method usually performs somewhere in between our adaptive beam search and regular beam search. There are a few exceptions: for some datasets (e.g., SIFT) it performs worse than beam search on some graphs (in contrast, our Adaptive Beam Search always performs better than standard beam search). We also note that, for MNIST, the Groh et. al method performs very close to our Adaptive Beam Search, which we attribute to the fact that the nearest-neighbor distance in this dataset is fairly uniform. In this case, a multiplicative vs. additive slack factor should behave fairly similarly. We will plan on adding these plots to the paper, as we believe they strengthen our work.
>
> Concerning a comparison to the method in [33], we have run experiments on the method in the past. Unfortunately, we found the method to be finicky and difficult to get good results on a variety of graphs and datasets. This seems to be due to a critical choice of hyperparameter that is difficult to estimate from the training data. We did not wish to include these results (which show [33] performing poorly in most scenarios) in case there is some implementation issue on our end. We would be happy to contact the authors to discuss the method so that we can include it for comparison. However, we do want to emphasize that we believe simple methods like standard beam search, Adaptive Beam Search, and the Groh et al. method are worth considering regardless of approaches like [33], which requires a training period and expects temporal consistency between queries to make the learned stopping rule meaningful in the future. In our experience, the simpler methods are more widely adopted in practice due to their robustness and ease of implementation.
>
> **W3:** The referenced Jiadong et al. paper came out after we made this submission. We did not ignore it – indeed, we have an email thread discussing the work! We are happy to continue considering that work, but do not have time to do a full experimental comparison during the short rebuttal period.
>
> **Q2:** Parameters for the graph construction methods are included in Appendix B, Table 3. Let us know if there are other parameters you are interested in.

---

> > ### Comment · Reviewer_ZV7M · 2025-08-04
> >
> > Thanks for the feedback from the authors. However, my concerns about this paper still exist.
> >
> > For the theoretical part, I would like to see more discussions on the differences between the slack factor based methods (both this work and the GGNN of  Fabian Groh et. al) and the adaptive graph structure with labeled/weighted monotonic graph (Jiadong Xie et.al TODS 2025). At least, those papers are publicly available.
> >
> > For the novelty part, the multiplicative slack factor is similar to the additive slack factor of GGNN, which weakens the novelty of the proposed method in this work, even considering the theoretical analysis on the multiplicative slack factor.
> >
> > For the experimental part, it is far from enough to reduce my concerns. Tables could be used for performance comparisons.
> >
> > Overall, even though I am not satisfied with the feedback, I will keep my rating unchanged.

---

> > > ### Author Response · Authors · 2025-08-05
> > >
> > > Regarding theoretical guarantees:
> > >
> > > The Jiadong Xie et. al paper builds on a paper title "Efficient Approximate Nearest Neighbor Search in Multi-dimensional Databases" by Peng et al. This paper introduces the concept of a $\tau$-monotonic graph, which is a strictly stronger property than navigability: for every points $i$ and $j$, $i$ must either be connected to $j$ directly, or be connected to some $k$ for which $d(k,j) < d(i,j) - 2\tau$. Interestingly, under this assumption, the authors prove that greedy search will return an *exact* nearest neighbor for any query whose nearest neighbor is within distance $\tau$. We plan to add further discussion of this result to our paper.
> > >
> > > Like the Indky-Xu work on $\alpha$-shortcut reachability, the above result is complimentary to ours: it provides a nice guarantee, but under a stronger assumption than we do. In particular, it is not hard to construct datasets where the sparsest $\tau$-monotonic graph has degree $\Omega(n)$ for any fixed $\tau$, but via reference [12], any dataset has a navigable graph with average degree $O(\sqrt{n})$. Moreover, like Indyk-Xu, the bounds for $\tau$-monotonic graphs hold even for vanilla greedy search, so unlike our work, the analysis does not capture the importance of relaxing greedy search in practice.
> > >
> > > Both the Peng et al. paper and the Jiadong Xie paper also provide additional runtime bounds under strong distributional assumption on the data (basically, uniformly distributed with sufficient density). In contrast, we are interested in bounds that hold for worst-case datasets.

---

> > > > ### Comment · Reviewer_ZV7M · 2025-08-06
> > > >
> > > > Thanks for the further explanation.
> > > >
> > > > The authors conducted an interesting theoretical analysis on ANNS based on the graph index. However, my main concerns still exist, i.e., (1) the limited novelty due to the similarity to the GGNN method and (2) no further comparisons with other methods of the same type, e.g., early termination methods via machine learning based methods and so on.
> > > >
> > > > Even though I am negative on this work overall, my rating of 3 is high in my view.

---

### Official Review · Reviewer_DYui · 2025-07-02

**Clarity:** 4
**Significance:** 4
**Originality:** 4
**Rating:** 5
**Confidence:** 5

**Summary:**

This paper presents a theoretical analysis and practical contributions regarding the search algorithm in graph-based approximate nearest neighbor search, known as beam search. The proposed method is called Adaptive Beam Search. Unlike conventional beam search, the proposed method fixes the beam width to match the top-k value while relaxing the distance condition related to the termination criterion of the algorithm. Controlling this relaxation parameter manages the trade-off between search speed and accuracy.

A notable aspect of the proposed method is that it provides a theoretical analysis of the beam search component, which has not been previously explored. Indeed, the existing literature contains few theoretical analyses of beam search itself, and more importantly, it lacks any discussion relating to the navigability of the graph. In this work, the conventional beam search is generalized through a highly intuitive concept of "termination condition", which leads to the proposed method. The resulting theoretical conclusions are intuitive. Moreover, in practice, the proposed method has been shown to outperform standard beam search across many datasets.

**Questions:**

The authors do not mention which libraries or implementations were used for the search method. It would be helpful to include such information. Did the authors implement everything from scratch, or did they build upon existing frameworks such as DiskANN or faiss?

Note that faiss is the de facto standard library for nearest neighbor search. If the proposed method can be integrated into faiss, users would welcome it.

**Ethical Concerns:**

["NO or VERY MINOR ethics concerns only"]

**Final Justification:**

The technical contribution of this paper is significant, and I will maintain my original accept score.

**Quality:**

4

**Strengths And Weaknesses:**

I enjoyed reading this paper. The raised problem and the proposed solutions are clear, and the paper significantly contributes to the nearest neighbor search field. Therefore, I evaluate this paper as an Accept.

# Strengths

The strengths of this paper are as follows:

## Theoretical contribution

The theoretical contribution regarding the beam search component addressed in this paper has rarely been explored in previous theoretical studies on graph-based nearest neighbor search. In particular, by focusing on the "termination condition", the authors generalize the existing beam search and conduct discussions within that framework. The theoretical result obtained (Theorem 1) is very intuitive and practical. It is expected that deeper theoretical analysis will be developed based on this work.

## Practical contribution

Although this is a theoretical paper, extensive experiments validated its practical value. Many theoretical papers on graph-based nearest neighbor search have been purely theoretical and often unrelated to actual search performance. This is unfortunate, because graph-based nearest neighbor search is popular and widely used precisely due to its high practical performance. Theoretical work should explain that performance, but instead, they have often become theory for the sake of theory. This paper, while theoretical, provides insights that are closely linked to the practical performance of graph-based methods.

## Completeness of the paper

Overall, the paper is very well-structured. In particular, it gives the impression of a well-written document in which the authors successively present questions or points that readers may be wondering about or sympathizing with, then provide answers to them in order.

# Weaknesses

I do not have any strong weaknesses to point out, but I offer a few comments below.

## Membership query for $\mathcal{D}$

In Line 7 of Algorithm 1, there is a statement "$y$ is not in $\mathcal{D}$." This description is slightly inaccurate when reading only Section 3.1 and Algorithm 1. This is because $\mathcal{D}$ is a priority queue, and checking for the existence of an element requires $O(|\mathcal{D}|)$ time. This point is clarified in Algorithm 2, which provides more detailed pseudocode and represents $\mathcal{D}$ as a dictionary, allowing for $O(1)$ membership checking. However, this is not evident from Algorithm 1 alone. A brief footnote in Section 3.1 mentioning this point may be helpful.

Also, in the faiss implementation of HNSW, a table (array) is used for $\mathcal{D}$. It might also be worth mentioning that different libraries adopt different implementations.

## V2 is necessary?

The only puzzling aspect throughout the paper is the introduction of Adaptive Beam Search V2 in Equation (6). I do not understand why this variant is introduced here. Its performance is inferior, and it does not need to appear in the main body of the paper. It would be better relegated to the supplementary material. Its presence slightly disrupts the rhythm of the paper overall.

---

> ### Author Rebuttal · Authors · 2025-07-30
>
> Thank you very much for the positive feedback. The comments on presentation are helpful, and we will include the reviewer’s suggestions in the next draft. We will also consider moving Adaptive Beam Search V2. Indeed, this method is a relic of explorations from early stages of this project. Given its poor performance, moving it to an appendix probably makes sense.
>
> In response to the question about implementation: all search algorithms were implemented from scratch. For the graph constructions (HNSW, NSG, Vamana, and EFANNA) some were implemented from scratch and some used existing codebases – see Appendix B.3 for details. Thanks for the suggestion about integrating the method into FAISS – this is definitely something we will consider.

---

> > ### Comment · Reviewer_DYui · 2025-08-03
> >
> > Thanks for the rebuttal. I maintain my original score.

---

### Official Review · Reviewer_gJ9t · 2025-07-03

**Clarity:** 4
**Significance:** 4
**Originality:** 2
**Rating:** 4
**Confidence:** 4

**Summary:**

This work provides a new stopping criterion (the eponymous adaptive beam search) for the popular beam search algorithm on graph-based approximate nearest neighbor search (ANNS) data structures. As long as the constructed graph is _navigable_, one can provably achieve a clean tradeoff between accuracy of the beam search and its speed (i.e. number of steps of the search before stopping). In addition, this work shows that this tradeoff doesn't exist for the standard beam search: in the navigability model, it's entirely possible for it to diverge entirely. In the empirical evaluation, graphs searched with the adaptive beam search use fewer distance comparisons than the standard beam search to achieve a fixed recall.

**Questions:**

1. In the $\alpha$-shortcut reachability setting, there does seem to be a recent paper on beam search (released publicly at ICML after the NeurIPS submission deadline): https://openreview.net/attachment?id=JnXbUKtLzz&name=pdf
If the authors believe the results to be analogous (with respect to $\alpha$-reachability and navigability), it would be nice to get their thoughts on how the key beam-search results compare.

2. With the weakness in mind, is it possible that the navigability property itself may be _too_ loose to provide a convergence guarantee? I'd appreciate the authors' thoughts on this, as the additional structure of the $\alpha$-shortcut reachability setup seems to be more amenable to such a guarantee (just looking at the main result of Indyk/Xu).

3. Similarly, is beam search's inability to converge for most possible beam-widths (claim 2) an issue with the navigability property? The paper linked above does seem to suggest that beam search does converge on graphs that satisfy $\alpha$-shortcut reachability.

4. Is it possible to get an implementation of the adaptive beam search with a single priority queue (like implementations of standard beam search)?

**Ethical Concerns:**

["NO or VERY MINOR ethics concerns only"]

**Final Justification:**

The authors provide an illuminating discussion comparing $\alpha$-shortcut reachability and navigability, and it helps in visualizing the difficulties that arise in producing (fast) convergence guarantees in the navigability model. However, I think this submission would have greatly benefitted from such a result, providing analogous results to existing work in the $\alpha$-shortcut reachability setting and complementing the empirical results.

Nevertheless, I believe the existing work is otherwise solid, and so I maintain my original score.

**Limitations:**

See questions, especially 3/4

**Quality:**

3

**Strengths And Weaknesses:**

**Strengths**

1. This contributions of this paper are primarily theoretical: by the admission of the authors, the stopping criteria specified in this work has been described as a heuristic in prior works. Contributing theoretical grounding of heuristics in a research area where theory is usually well behind the empirical state-of-the-art is welcome and necessary.

2. Navigability (in comparison to $\alpha$-shortcut reachability) is a simple requirement that can (approximately) model a number of popular graph datastructures used in practice (HNSW, DiskANN, etc).

3. The paper is well-written and easy to digest.

4. The empirical evaluation is thorough and well-documented, and nicely complements the theoretical results.


**Weaknesses/Limitations**
While this work does show that adaptive beam search arrives at an approximate solution parametrized by $\gamma$, it does not show any convergence guarantees: it might take an inordinate number of steps through the graph to achieve the result stated in theorem 1, although the empirical study seems to disagree with that possibility.

---

> ### Author Rebuttal · Authors · 2025-07-30
>
> Thank you for the positive feedback and useful questions. We address the specific questions below:
>
> **Question 1.** We also became aware of this paper after our submission, and plan to add a citation. Basically, the paper generalizes the cited work of [Indyk, Xu, NeurIPS 2023] to approximate $k$-nearest neighbors for $k > 1$. What is called “beam search” in that paper is equivalent to what we call “classic greedy search”, since the beam width is set exactly equal to $k$, the number of nearest neighbors desired.
>
> We believe that, like the Indyk-Xu work, the referenced work is complementary to ours. It also provides a multiplicative error guarantee (as well as a stronger convergence time bound). However, the bounds in these papers apply only when G is $\alpha$-shortcut reachable for $\alpha$ strictly greater than 1. In particular, their error bound is of the form $(\alpha+1)/(\alpha-1)$ for general metrics, so it becomes vacuous when $\alpha = 1$, which corresponds to the navigability property. In general, $\alpha$-reachability is a much stronger property than navigability when $\alpha > 1$: as discussed in Section 2, navigable graphs with sublinear sparsity have been shown to exist for any dataset, whereas it is easy to construct datasets where any $\alpha$-reachable graph requires $O(n^2)$ edges, and thus does not admit fast search, even if the search converges quickly – see our Appendix A.2 for more detail. We thus think it is valuable to study guarantees for approximate neighbor search under the weaker notion of navigability.
>
> Perhaps as further reason to study navigability, recent work by Conway et al. “Efficiently Constructing Sparse Navigable Graphs” and Khanna et al. “Sparse Navigable Graphs for Nearest Neighbor Search: Algorithms and Hardness” both study algorithms for finding minimum sparsity navigable and $\alpha$-shortcut reachable graphs. Based on the results of these papers, it seems that finding a minimum sparsity navigable graph for a given point set is an easier problem than finding a minimum sparsity $\alpha$-shortcut reachable graph.
>
> Finally, we find it interesting that our bounds require some sort of relaxation on greedy search. As mentioned, the referenced work and that of Indyk-Xu both obtain the same guarantees even when running beam search with $b = k$ (i.e., running classical greedy search). While very nice results, these bounds thus do not provide theoretical evidence for why a choice of beam width larger than $k$ (or, in our case, $\gamma > 0$) is critical to good performance in practice.
>
> **Question 2.** Strictly speaking, navigability itself is indeed too weak a notion to prove convergence guarantees. Consider e.g., $n$ points arranged on a line, connected with a bi-directional path. This is a navigable graph, but if the query is on one end of the line and the starting vertex is on the other, convergence may require $n$ steps.
>
> Nevertheless, citation [12] in the paper shows that it is possible to construct sparse navigable graphs with “depth 2”, meaning that greedy search converges in just two steps if the query is in the dataset. We agree that a very interesting future direction would be to see if properties like low-depth can be leveraged to prove fast convergence for queries not in the dataset (for which the goal is to return an approximate solution). We do not see any inherent barriers to doing so.
>
> **Question 3.** Yes, you are correct. The cited result proves convergence for standard beam search for $\alpha$-shortcut reachable graphs. Our perspective on this is as follows: if you want the flexibility to work with navigable graphs (which are in general sparser and easier to find) you need to look beyond standard beam search. As mentioned in the response to Question 1, $\alpha$-shortcut reachability seems to be stronger than what is achieved by heuristic graph constructions used in practice – indeed for these constructions, beam width $> k$ is generally required for good performance. However, the cited result proves convergence results for $\alpha$-shortcut reachable graphs even for beam width $k$ (i.e., classic greedy search).
>
> **Question 4.** This is a good question – we do not immediately see how to do this. In our implementation, however, data structures only factored minimally into total runtime, which was dominated by distance/inner product computations. Note that the second priority queue ($\mathcal{B}$ in Algorithm 2) has size bounded by $k$, the number of target nearest neighbors. Since $k$ is typically rather small, the data structure is inexpensive to maintain.

---

### Decision · Program_Chairs · 2025-09-17

**Decision:**

Accept (poster)

**Comment:**

The paper gives theoretical analysis for the beam search component of graph based nearest neighbor search indices. The theoretical insights give a better empirical algorithm which outperforms standard beam search across many datasets. The reviewers also highlighted the readability and clarity of the paper. This paper contributes towards a growing body of work understanding graph based data structures for search and I believe the insights will lead to further downstream work.